



**Altimetry, gravimetry, GPS and viscoelastic modelling data for the joint inversion for**

**glacial isostatic adjustment in Antarctica (ESA STSE Project REGINA)**

Ingo Sasgen[1], Alba Martín-Español[2], Alexander Horvath[3], Volker Klemann[4], Elizabeth J.

Petrie[5], Bert Wouters[6], Martin Horwath[7], Roland Pail[3], Jonathan L. Bamber[2], Peter J.

6      Clarke[8], Hannes Konrad[9], Terry Wilson[10] and Mark R. Drinkwater[11]

1.  Division of Climate Sciences, Alfred Wegener Institute, Bussestraße 24, 27570

9      Bremerhaven, Germany.

2.  School of Geographical Sciences, University of Bristol, University Road, Clifton,

Bristol BS8 1SS, United Kingdom.

12   3.  Institut für Astronomische und Physikalische Geodäsie, Technische Universität

München, Arcisstraße 21, 80333 München, Germany.

4.  Department of Geodesy, GFZ German Research Centre for Geosciences,

15      Telegrafenberg, 14473 Potsdam, Germany.

5.  School of Geographical and Earth Sciences, University of Glasgow, Glasgow, G12

8QQ, United Kingdom.

18   6.  Institute for Marine and Atmospheric Research, Utrecht University, Princetonplein 5,

3584 CC, Utrecht, The Netherlands.

7.  Institut für Planetare Geodäsie, Technische Universität Dresden, Helmholtzstr. 10,

21      01069 Dresden, Germany.

8.  School of Civil Engineering and Geosciences, Newcastle University, Newcastle, NE1

7RU, United Kingdom.



24    9. School of Earth and Environment, University of Leeds, Leeds, LS2 9JT, United

Kingdom.

10. School of Earth Science, Ohio State University, 275 Mendenhall Lab, 125 South Oval

Mall, Columbus OH, 43210, USA.

11. Mission Science Division, European Space Agency, European Space Research and

Technology Centre, Keplerlaan 1, Noordwijk 2201 AZ, The Netherlands.

**Keywords:**

Global change from geodesy, Gravity anomalies and Earth structure, Loading of the Earth,

Glaciology, Antarctica, Joint inversion



**ABSTRACT**

A major uncertainty in determining the mass balance of the Antarctic ice sheet from measurements of satellite gravimetry, and to a lesser extent satellite altimetry, is the poorly known correction for the ongoing deformation of the solid Earth caused by glacial isostatic

adjustment (GIA). In the past decade, much progress has been made in consistently modelling the ice sheet and solid Earth interactions; however, forward-modelling solutions of GIA in Antarctica remain uncertain due to the sparsity of constraints on the ice sheet evolution, as well

as the Earth's rheological properties. An alternative approach towards estimating GIA is the joint inversion of multiple satellite data – namely, satellite gravimetry, satellite altimetry and GPS, which reflect, with different sensitivities, trends of recent glacial changes and GIA.

Crucial to the success of this approach is the accuracy of the space-geodetic data sets. Here, we present reprocessed rates of surface-ice elevation change (Envisat/ICESat; 2003-2009), gravity field change (GRACE; 2003-2009) and bedrock uplift (GPS; 1995-2013.7). The data analysis

is complemented by the forward-modelling of viscoelastic response functions to disc load forcing, allowing us to relate GIA-induced surface displacements with gravity changes for different rheological parameters of the solid Earth. The data and modelling results presented

here are available in the Pangeae archive; https://doi.pangaea.de/10.1594/PANGAEA.875745. The data sets are the input streams for the joint inversion estimate of present-day ice-mass change and GIA, focusing on Antarctica. However, the methods, code and data provided in this

paper are applicable to solve other problems, such as volume balances of the Antarctic ice sheet, or to other geographical regions, in the case of the viscoelastic response functions. This paper presents the first of two contributions summarizing the work carried out within a European

Space Agency funded study, REGINA.



**COPYRIGHT STATEMENT**

The work presented here is provided under the terms of the Creative Commons License

Attribution 3.0 Unported (CC BY 3.0).



## 1. INTRODUCTION

Glacial isostatic adjustment (GIA), the viscoelastic deformation of the solid Earth in response to climate-driven ice and water mass redistribution on its surface, is poorly constrained in Antarctica. The primary reason is the sparseness of geological evidence of the past ice sheet geometry and local relative sea-level change. These are important constraints on the exerted glacial forcing and on the viscoelastic structure of the lithosphere and of the mantle, respectively, which concertedly determine the signature of GIA (e.g. Peltier, 2004; Ivins and James 2005; Whitehouse et al. 2012: van der Wal et al., 2015). The predictions of GIA in Antarctica remain ambiguous (Shepherd et al. 2012, suppl.) and cause a large uncertainty in gravimetric mass balance estimates of the ice sheet of the order of the estimate itself (Martín-Español et al. 2016b). Measurements of bedrock uplift by GPS have shown to be inconsistent with forward models, which tend to over-predict uplift and mass increase due to GIA, biasing estimates of present-day Antarctic ice-mass loss from GRACE to more negative values (Bevis et al. 2009).

Much progress has been made in reconstructing the ice sheet evolution from geomorphological evidence (Bentley et al. 2014) and inferring the underlying Earth structure from seismic observations (An et al. 2015; Heeszel et al. 2016). However, an independent approach to constraining GIA is to make use of the different sensitivities of the various types of satellite data to recent glacial changes and GIA, respectively. And thus to separate both signals in a joint inversion approach has been pursued by e.g. Wahr et al. 2000; Riva et al. 2009; Wu et al. 2010; Gunter et al. 2014, Martín-Español et al. 2016a. Another approach used regional patterns of GIA from forward modelling and adjusted them to GIA uplift rates in Antarctica (Sasgen et al. 2013).

In this paper, we present methods and data inputs in preparation of solving the joint

inversion for GIA in Antarctica. As the GIA process is gradual, causing an approximately

constant rate of change within a decade, we first process the satellite data to recover optimal

temporal linear trends. We refine existing procedures for the surface-ice elevation changes from

Envisat and ICESat satellite altimetry (Section 2), bedrock displacement from *in situ* networks

of GPS stations in Antarctica (Section 3), and gravity field change from GRACE (Section 4).

We also present forward modelling results of viscoelastic response functions to disc load forcing

for the range of Earth structures likely to prevail in Antarctica (Section 5).

The determination of viscoelastic response functions is a classic topic in solid Earth

modelling (e.g. Peltier & Andrews, 1976), though uncommon the application to joint invasion

studies of satellite data. Although this paper focusses on Antarctica, the response functions and

data processing techniques presented here are applicable to other regions. The response kernels

represent a wide range of Earth structures and can be used for the separation of superimposed

present-day (elastic) and past (viscoelastic) signatures of mass change in other regions, for

example hydrological storage changes and GIA in North America. The response functions give

insight into the temporal and spatial scales of deformation expected for Antarctica, and are

crucial when combining the input data streams.

The data sets and modelling results presented in this paper are accessible in the Pangeae

archive, https://www.pangaea.de/ – subsections provide user guidance and point to data and

code stored in the archive. As mentioned above, the data sets and modelling results are of value

to address other research questions as well. For example, the GPS rates provided are useful for

the validation of forward modelling GIA solutions, the GRACE gravity rates can be used for

mass balance studies, and altimetry data 2003-2009 can be extended with the ongoing CryoSat-



2 mission to infer volumetric mass balances, also over the ice shelves. The viscoelastic response

functions are based on Earth model parameters suitable to other geographical regions, as well;

they are useful for similar studies combining different data sets of geodetic observables, surface

deformation, gravity field change, and topographic change in glaciated areas.

The actual method of the joint inversion is described in a second contribution of the

REGINA project team (Sasgen et al. *submitted*). In this second paper, the resulting GIA estimate

is also compared to previous studies. The processing of the data issued here was enabled by the

European Space Agency within the CryoSat+ Support To Science Element Study REGINA.



## 2. ALTIMETRY DATA ANALYSIS

### 2.1 *ICESat elevation rate determination*

We use along-track altimetry measurements from *ICESat 633 Level 2*, providing high-resolution elevation change observations for the period February 2003 until October 2009. Two corrections are applied to this data set: the range determination from Transmit-Pulse Reference-Point Selection (Centroid vs. Gaussian) (Borsa et al. 2014) available from the National Snow and Ice Data Center (NSIDC), and the inter-campaign correction (Hofton et al. 2013). The Centroid-Gaussian correction is a well-established correction and has been incorporated to the latest ICESat release (634). Concerning the ICESat Intercampaign Bias (ICB) correction, uncertainties are available at Hofton et al (2013). Furthermore, several studies have determined this correction from different methodologies. For a summary of published ICESat ICB corrections see Scambos & Shumman (2016). Because ICESat tracks do not usually overlap, a regression approach is used in which topographic slope (both across-track and along-track) and the rate of surface-elevation change $y^h_{\mathrm{ICESat}}$, are simultaneously estimated using the 'plane' method (Howat et al. 2008´) over areas spanning 700 m long and few hundred meters wide. A regression is only performed if a plane has at least 10 points from four different tracks that span at least one year. Regression was carried out twice; first, individual elevation measurements with corresponding residuals outside the range of two standard deviations were detected, then, the regression was repeated omitting these outliers. The standard deviation of the regression coefficient, here taken as the uncertainty of the elevation rate, $\sigma^h$(here, ICESat) is calculated by the propagation of the residual uncertainties of the topographic heights,

$$\hat{s}_{\mathrm{ICESat}} = \sqrt{\frac{\sum e_i^2 / (n-2)}{\sum (x_i - \bar{x})^2}} \ , \ (1)$$



to the trend parameter, where $e$ is the vector of residuals, $n$ is the sample size ($i = 1, 2, \ldots, n$), and $x$ is the vector of input elevations with mean $\bar{x}$. This standard deviation ($\sigma_{\mathrm{ICESat}}$) takes into account the sample size and the variance of both input data and residuals of the regression (Hurkmans et al. 2012). The exact ICESat observation periods are shown in the Appendix (A.1, Table A.1). Then, the elevation rate and its uncertainty are interpolated to a common $10 \times 10$ km grid in polar-stereographic projection (central latitude 71°S; central longitude 0°W, and origin at the South Pole, WGS-84 reference ellipsoid).

### 2.2 *Envisat elevation rate determination*

We use a time series of elevation changes derived from along-track Envisat radar altimetry data for the interval January 2003 to October 2009 (coeval to ICESat time span). Elevation rates $y^h_{\mathrm{Envisat}}$ are obtained at points every 1 km along track, by binning all the echoes within a 500 m radius. Then, a 10-parameter least squares model is fitted in order to correct for the across-track topography and changes in snowpack properties. The least square model is defined in Flament and Remy (2012). The estimated parameters include parameters determined for the backscatter, leading-edge width and tailing-edge slope, the mean altitude, quadratic surface slope parameters to define surface curvature and a linear time trend. A digital elevation model was not used for the correction of the topographic slope. For processing reasons, the temporal resolution is re-sampled from 35 days to monthly periods for each grid cell, before estimating the elevation rates. This has a minor effect on the elevation rate estimate (smaller than $\pm$ 1 cm) and reduces the standard deviation by about 14 %. As for ICESat, the elevation rate is interpolated to a common $10 \times 10$ km polar stereographic grid, and the standard deviations of the rates within each grid cell are taken as an estimate of the measurement uncertainty, $\sigma_{\mathrm{Envisat}}$.



2.3 *Combination of Envisat and ICESat*

We produce a combined rate of surface-elevation change product from the ICESat and

Envisat datasets for the Antarctic ice sheet, $y^h$. The aim is to take advantage of the high spatial

resolution of ICESat data and the high temporal resolution and high-track density of the Envisat

data.

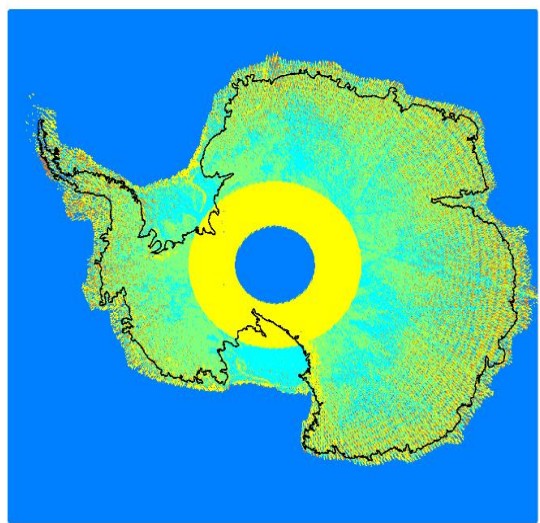

*Figure 1. Mask for the combination of Envisat/ICESat. ICESat but not Envisat available (yellow),*

*$\sigma_{ICESat} \leq \sigma_{Envisat}$ (green), $\sigma_{ICESat} > \sigma_{Envisat}$ (turquoise), Envisat but not ICESat*

*available (orange), and no data (blue). No interpolation is used.*

We combine the two altimetry datasets based on their common $10 \times 10$ km polar-

stereographic grid. At each location, the elevation rate with the smallest standard deviation is

chosen from either Envisat or ICESat datasets.

Fig. 1 shows the resulting mask underlying the combination. It is evident that some grid

points are only represented by either ICESat or Envisat. Most prominent is the narrowing of the

polar gap with ICESat data, resulting from the 81.5°S latitude limit for Envisat compared to



86°S for ICESat due to satellite orbit inclination. On the Antarctic Peninsula, Envisat picks up

some points that are not present due to a sparser track coverage in the ICESat data set. As

expected, ICESat outperforms Envisat in terms of uncertainty of the elevation rate over steep

topographic slopes and along the ice sheet margins. This is due to the smaller footprint of the

laser altimeter, its higher accuracy and lower slope-dependent uncertainty (e.g. Brenner 2007).

On some flat areas and over some faulty ground tracks, where ICESat data measurements are

scarce, however, Envisat provides better temporal and spatial coverage leading to better

accuracy of the resulting elevation rates. The resulting combined data set of surface-elevation

rates and its uncertainties are shown in Fig. 2.





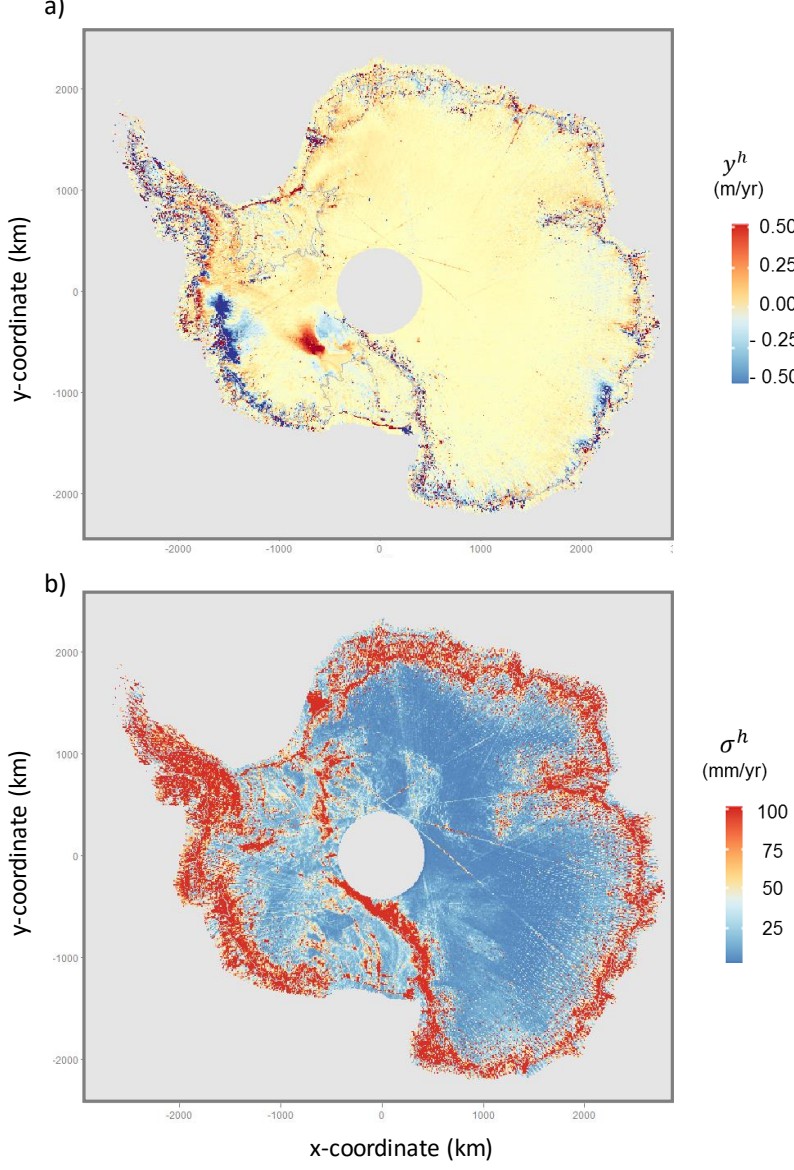

Figure 2: a) Rate of surface-ice elevation change $y_h$ and b) associated uncertainties $\sigma_h$ derived

from Envisat/ICESat combined dataset for the time interval 2003-2009. No interpolation is

used; grid points without values are empty (shaded grey).



### 2.4 *Firn correction*

The elevation rates derived from ICESat and Envisat are corrected for changes in the firn

layer thickness using the firn compaction model of Ligtenberg (2011), which is driven by the

regional atmosphere and climate model RACMO2/ANT (Lenaerts, 2010). We determine the

firn compaction for January 2003 to October 2009, with respect to the mean of the years 1979

to 2002 and estimate a temporal linear trend, $h_{comp}$. The model output is re-gridded onto the

$10 \times 10$ km common grid using nearest neighbor interpolation. The standard deviation of the

re-gridding is less than 1 cm/yr, causing a maximum change of 2 % of the firn compaction rate.

Note that the firn compaction model has a spatial resolution of 27 km, potentially neglecting

finer-scale processes relevant for the altimetry data. Clearly, the re-gridding uncertainty stated

above is merely a minimum estimate, neglecting, for example, uncertainties in the calibration

or the atmospheric forcing of the firn compaction model.

The data were re-sampled from every two days to monthly mean time periods for every

grid cell before estimating elevation rates. As for the Envisat and ICESat data, no seasonal terms

are co-estimated and removed (i.e. annual and semi-annual). We do not apply an *a priori*

correction for surface-mass balance (SMB) trends, in accordance with the GRACE processing

(Section 5), which requires defining a climatological reference period. Note that applying the

commonly used reference period (1979 to present) leads to spurious accumulation anomalies

in the altimetry data (see Appendix A.2, Fig. A.1). The derivation of an adequate climatological

reference epoch in the RACMO2/ANT simulations is in itself challenging and beyond the scope

of this paper.

The total uncertainty of the rate of elevation change from satellite altimetry is calculated

by



$$\sigma_{\mathrm{h}} = \sqrt{\sigma_{\mathrm{Envisat/ICESat}}^2 + \sigma_{\mathrm{Firn}}^2} \, , (2)$$

where the standard deviation of the firn correction, $\sigma_{\mathrm{Firn}}$ is the formal regression

uncertainty (neglecting model uncertainties, as these are not available), and we assume the error

sources to be uncorrelated.

2.5 *Data availability*

Annual elevation trends from a combination of Envisat and ICESat data for the time period

between February 2003 and October 2009. Trends have been corrected for firn densification

processes using RACMO2/ANT. Elevation trends are provided in a 20 km polar stereographic

grid (central meridian 0∘ , standard parallel 71∘ S) with respect to the WGS84 geoid. X and Y

are given in km, and the elevation rate and its standard deviation are given in m/yr.

2.5.1 *ICESat elevation trend for the* time period between February 2003 and October

     2009.

The dataset is provided in a 10 km grid in polar stereographic projection (central meridian

0∘da standard parallel 71∘ S) with respect to the WGS84 geoid. X and Y are given in km, and

the elevation rate and its standard deviation are given in m/yr.

     2.5.2 *Envisat elevation trend for the time period between February 2003 and October*

*2009.*

The dataset is provided in a 10 km grid in polar stereographic projection (central meridian

0∘ , standard parallel 71∘ S) with respect to the WGS84 geoid. X and Y are given in km, and the

elevation rate and its standard deviation are given in m/yr.



### 2.5.3 ICESat & Envisat combination for time period between February 2003 and October 2009.

Elevation changes have been corrected for firn densification processes using a FDM. The dataset is provided in a 10 km grid in polar stereographic projection (central meridian 0∘fo standard parallel 71∘ S) with respect to the WGS84 geoid. X and Y are given in km, and the

elevation rate and its standard deviation are given in m/yr.

### 2.5.4 *Annual elevation trends from CryoSat-2 derived from a single trend covering the time period 2010-2013.*

An acceleration term in areas with dynamic thinning was added to the linear trend to obtain annual rates. Elevation trends are provided at 10 km resolution in a polar stereographic grid (central meridian 0∘ , standard parallel 71∘ S) with respect to the WGS84 geoid. X and Y are

given in km and the elevation rate and its standard deviation are given in m/yr.

### 2.5.5 *Elevation changes from firn model*

Annual firn densification rates over 2003-2013 rates obtained from RACMO2.3. Data is

240 provided in a 27 km polar stereographic grid (central meridian 0∘ , standard parallel 71∘ S) with respect to the WGS84 geoid. X and Y are given in km and the annual firn densification rates in m/yr.

### 2.5.6 *Snow / ice density map*

The density map for volume-to-mass conversion is provided in 20 km resolution in a polar stereographic grid (central meridian 0∘ , standard parallel 71∘ S) with respect to the WGS84

geoid. X and Y are given in km and density in km/m³.

### 2.5.7 *ICESat/Envisat combination mask*

Mask used for combining ICESat and Envisat in a 10 km resolution and polar stereographic

coordinates.

X and Y are coordinates in km and the id represents whether ICESat or Envisat has been

used to construct the elevation change combination.

4: only Envisat was available

3: only ICESat was available

2: ICESat lower errors

1: Envisat lower errors

### 3. GPS UPLIFT RATE ESTIMATION & CLUSTERING

The aim of the GPS time series analysis is to derive uplift rates, $y_u$ that represent the

geophysical ground motion at the sites as accurately and robustly as possible. We derive uplift

rates based on GPS records from a total of 118 Antarctic sites. Data were processed from 1995

day of year (doy) 002 to 2013 doy 257 (1995.0-2013.7) but data at individual sites are of varying

length and quality. The processing and uplift rate and uncertainty estimation methodology are

documented in detail in Petrie et al. (in prep. a, b), but a short summary is given here for

convenience. It resembles that of Thomas et al. (2011), but with more recent processing

software (GIPSY 6.2) and model updates (including second order ionospheric and earth

radiation models): an initial satellite orbit and clock estimation step is performed, using a

carefully selected balanced stable global network of GPS sites. The orbits and clocks are then

used to perform precise point positioning (PPP) processing of all the available Antarctic sites

of interest. A mini-ensemble was created to investigate systematic processing uncertainties and

manual investigation was performed of effects of possible systematic errors in the time series

on uplift rates. The mini-ensemble investigation showed that decisions taken when analyzing

time series tended to have larger effects on uplift rates and uncertainties than the effects of small

processing strategy changes. Outliers and systematic errors, such as offsets due to equipment

changes or other causes, were removed where possible. Due to the varying characteristics of

the time series it was not possible to use the same approach at all sites. The strategy was as

follows (and is summarized in Appendix A.3, Fig. A.3). For sites with over 2000 days of data,

uplift rates and associated uncertainties were estimated using the CATS software (Williams

2008). We co-estimated a white-noise scale factor for the formal uncertainties, and a power-law

noise amplitude with the index fixed to -1 (flicker noise), along with the temporal linear trend

(rate), seasonal (annual and semi-annual) parameters, and sizes of the offsets (at the specified

epochs).

     The median values of the white-noise scale factor and the power law noise amplitude,

derived from these long time series, were then used to propagate rates and uncertainties for the

shorter time series, for which CATS cannot produce reliable estimates. For the propagation, the

time series with fewer than 2000 epochs are additionally subdivided into two categories;

continuous sites ($\geq 2.5$ yr), for which periodic parameters are estimated in the propagation of

uncertainties, and very short continuous sites ($< 2.5$ yr) and campaign sites for which periodic

parameters are not estimated. For each campaign, 1 mm of noise was added when propagating

the uncertainties, to allow for tiny differences when re-setting up equipment.

     Finally, for each site, the uplift rate $y^u$ and its uncertainty $\sigma^u$ are assessed by manually

removing portions of the time series (for example deleting campaigns in turn). If the rate

changes by an amount larger than the propagated uncertainty for the site, the uncertainty is

assigned as $\pm$ the maximum difference in rate, and the rate is adjusted, if necessary, to the values

of the most likely part of the range. Sites with only two campaigns were assigned an uncertainty

of $\pm$ 100 mm/yr, unless there was further evidence for or against the existence of systematic

errors.

Table 1 summarizes the rate estimation methods and the number of sites for each. For

further details and full information on individual rates and time series, see Petrie et al. (in prep

a) for a full description of the processing and ensemble evaluation, and Petrie et al. (in prep b)

for details of time series analysis and rate and uncertainty estimation. Table 1 shows the

numbers of sites at which each approach was taken. Further work was undertaken to combine

or 'cluster' the rates regionally for inclusion in the estimation process – see the REGINA Paper

II (Sasgen et al. *submitted.)* for details.



*Table 1: Number of sites for each GPS uplift rate and uncertainty estimation method.*

| Rate and uncertainty estimation method | Number of sites (118 total) |
|---|---|
| CATS rate and uncertainty  ('cats, cats') | 18 |
| CATS rate, manually increased uncertainty ('cats, eman') | 2 |
| Propagated rate and uncertainty  ('prop, prop') | 28 |
| Propagated rate and manually increased uncertainty ('prop, eman') | 50 |
| Manually adjusted rate and manually increased uncertainty ('rman, eman') | 20 |

*Table 2. Uplift rates $y^u$ and associated uncertainties $\sigma^u$ (mm/yr) for selected GPS sites with more than 2000 epochs of data, compared to data published by Thomas et al. (2011) and Argus et al. (2014). Temporal components and noise characteristics are derived using the CATS software (Williams 2008), i.e. 'cats, cats' method.*

| Site | REGINA | | Thomas et al. (2011) | | Argus et al. (2014) | |
|---|---|---|---|---|---|---|
| | $y^u$ | $\sigma^u$ | $y^u$ | $\sigma^u$ | $y^u$ | $\sigma^u$ |
| cas1 | **1.5** | 0.2 | **1.2** | 0.4 | **1.7** | 0.8 |
| crar | **0.7** | 0.4 | **1.0** | 0.7 | **1.0** | 0.6 |
| dum1 | **-0.3** | 0.3 | **-0.8** | 0.5 | **−0.2** | 0.8 |
| maw1 | **-0.4** | 0.2 | **0.1** | 0.4 | **0.2** | 0.6 |
| mcm4 | **0.8** | 0.2 | **0.7** | 0.4 | | |
| sctb | **0.9** | 0.5 | **0.6** | 1.1 | | |
| syog | **1.1** | 0.2 | **2.3** | 0.4 | **0.6** | 0.8 |
| tnb1 | **0.1** | 0.5 | **-0.2** | 0.8 | **−0.4** | 1.0 |
| vesl | **0.4** | 0.3 | **1.1** | 0.5 | **1.5** | 0.8 |
| McMurdo* | | | | | 1.0 | 0.6 |

*Sites: crar-sctb-mcm4-mcmd

### 3.1 *Comparison with existing results*

Next, we briefly compare the uplift rates at individual sites (data span 1995.0-2013.7)



derived from the GPS processing described above with those available from three previous

studies: Thomas et al. (2011) (data span 1995.0-2011.0), Argus et al. (2014) (data span 1994-

2012) and the more geographically limited set of Wolstencroft et al. (2015) (data span 2006-

late 2013, focused on Palmer Land). It should be noted that the REGINA and Wolstencroft et

al. (2015) rates are in ITRF2008, the Thomas et al. (2011) rates are in ITRF2005 (which has

negligible scale or translation differences to ITRF2008), and the Argus et al. (2014) rates are in

a reference frame specific to the paper which they note yields 0.5 mm/yr more uplift than

ITRF2008 at high southern latitudes.

Due to the large number of Antarctic sites, in total 118, we focus the comparison on the

*Table 3. Uplift rates $y^u$ and associated uncertainties $\sigma^u$ (mm/yr) for selected GPS sites with fewer than 2000 epochs for data, compared to data published by Thomas et al. (2011) and Argus et al. (2014). Noise characteristics are derived median values from CATS software results for longer station records and propagated in the parameter estimation ('prop, prop' method). See Appendix A.4, Table A.2 for a full list of rates from this study.*

| Site | REGINA | | Thomas et al. (2011) | | Argus et al. (2014) | |
|---|---|---|---|---|---|---|
| | $y^u$ | $\sigma^u$ | $y^u$ | $\sigma^u$ | $y^u$ | $\sigma^u$ |
| belg | **-1.4** | 0.7 | **3.0** | 1.5 | **0.8** | 2.4 |
| dupt | **11.5** | 1.1 | | | **12.4** | 2.5 |
| fonp | **13.5** | 1.8 | | | **14.8** | 3.4 |
| frei | **-4.4** | 0.7 | | | **−2.9** | 1.4 |
| hugo | **0.9** | 1.3 | | | **1.7** | 3.6 |
| robi | **8.7** | 1.5 | | | **8.7** | 3.2 |
| roth | **5.5** | 1.4 | | | **5.4** | 1.4 |
| svea | **1.3** | 1.1 | **2.1** | 2.0 | **1.7** | 2.9 |
| vnad | **4.4** | 1.1 | | | **5.2** | 2.5 |

uplift rates and uncertainties derived by the methods 'cats, cats' (Table 2) and 'prop, prop'

   (Table 3). Uplift rates resulting from our study are provided in Appendix A.4 for all sites (Table

   A.2). Tables A.3 shows comparisons with the values of Thomas et al. (2011) and Argus et al.

(2014) for 'prop, eman' sites not shown in the main text. All uplift rates, $y^u$, are in mm/yr,

   with uncertainties reflecting 1-sigma standard deviations, $\sigma^u$. Sites with particularly complex

   non-linear time series such as those at O'Higgins (ohi2, ohig) and Palmer (palm) in the Antarctic

Peninsula are omitted here, as comparison with different studies is potentially misleading due

   to the effects of different measurement time periods. Table 2 shows data for selected sites with

   long time series, where uplift rate and uncertainty were derived using the CATS software

(Williams 2008). Uplift rates at the majority of the GPS sites agree within uncertainty, except

   syog (Syowa), where the REGINA value is between that from the other two studies. The

   uncertainty limits for the REGINA value and the Argus et al. (2014) just meet at 0.9 mm/yr,

even when allowing for the ~0.5 mm difference in reference frames, but the Thomas et al.

   (2011) value does not. This may be due to the fact that Thomas et al. (2011) estimate two offsets

   in the series. Table 3 shows uplift rate comparisons for sites where the 'prop,prop' method was

330 used; the noise characteristics are derived from median values from CATS software results for

   longer site records and then propagated in the parameter estimation in which annual and semi-

   annual parameters were also estimated along with the trend. Again, the rates agree within

333 uncertainty, except for site belg where there is a disagreement with Thomas et al. (2011). This

   may be due to their shorter data span. Table 4 shows comparisons for sites where the REGINA

   rates and uncertainties have been manually evaluated based on the spread of rates obtained by

336 sub-sampling the time series ('rman' method). There is a large difference (over 10 mm/yr) in

   the values at capf (Cape Framnes) between the REGINA value (4.0 ± 1.4 mm/yr) and the Argus



et al. (2014) value (15.0 ± 4.2 mm/yr). Interestingly, the Wolstencroft et al. (2015) rate values

for bean, gmez, lntk, mkib, and trve are all systematically higher than the REGINA values, by

an average of just over 3 mm/yr, and the uncertainties we assigned are also several times larger.

For more detailed analysis of rates and time series at individual sites, see Petrie et al. (in prep

b).

*Table 4. Uplift rates $y^u$ and associated uncertainties $\sigma^u$ (mm/yr) for selected sites where uplift rates are manually evaluated based on the spread of rates obtained by sub-sampling the time series ('rman' method), compared to data published by Thomas et al. (2011), Argus et al. (2014), Wolstencroft et al. (2015). See also 'rman' sites in Table Appendix A.4, Table A.2.*

| Site | REGINA | | Thomas et al. (2011) | | Argus et al. (2014) | | Wolstencroft et al. (2015 ) | |
|------|--------|--------|------|--------|------|--------|------|--------|
| | $y^u$ | $\sigma^u$ | $y^u$ | $\sigma^u$ | $y^u$ | $\sigma^u$ | $y^u$ | $\sigma^u$ |
| bren | **3.1** | 1.1 | **3.9** | 1.6 | **2.1** | 3.7 | **3.2** | 0.8 |
| capf | **4.0** | 1.4 | | | **15.0** | 4.2 | | |
| dav1 | **-1.6** | 0.6 | **-0.9** | 0.5 | **−0.8** | 1.0 | | |
| mait | **0.4** | 1.1 | **0.1** | 0.6 | **1.3** | 0.7 | | |
| mbl3 | **1.3** | 17.9 | **0.1** | 2.0 | | | | |
| bean | **2.1** | 4.3 | | | | | **7.5** | 1.2 |
| gmez | **1.5** | 4.8 | | | | | **5.7** | 0.8 |
| lntk | **4.6** | 3.1 | | | | | **6.0** | 0.7 |
| mkib | **4.7** | 2.6 | | | | | **6.9** | 0.5 |
| trve | **2.5** | 5.6 | | | | | **4.7** | 0.6 |

3.2 *Data availability*

3.2.1  *Bedrock uplift rates*

Bedrock uplift rates derived for the REGINA project are available in the text file

"REGINA_rates_full.txt", as presented in Table A.2 and A.3 of the Appendix A.4. The files

"REGINA_rates_03-13.txt" and "REGINA_rates_03-09.txt" contain subsets of the data, with

the temporal coverage limited to 2003-2013.5 and 2003-2009, respectively. The files are

organized as follows:

Lon [°], Lat [°], uplift rate [mm/yr], uncertainty of the uplift rate [mm/yr], GPS site ID

These *.txt files are the input to the clustering script described below. No elastic correction

has been applied.

3.2.2  *Clustering script*

In addition to the uplift rates for individual GPS sites, we provide a *bash* script "cluster.sh"

for clustering the heterogeneous data according to their geographic locations, for a pre-defined

threshold value. The idea is to reduce stochastic and geophysical noise of neighboring stations

in order to obtain uplift rates that are better regional representations for the length scale

recovered with GRACE (ca. 200 km). In an iterative procedure, the script selects neighboring

sites within a threshold and calculates the weighted average of the uplift rates and a simple

uplift of the stations locations. Input to the script are the REGINA rate files, specified in the

previous Section 3.2.1. Further details can be found in REGINA paper II (Sasgen et al.,

submitted). Note that the script relies on the open-source program suite Generic Mapping Tools,

http://gmt.soest.hawaii.edu/  (Wessel et al. 2013). Similar clustering can be achieved with the

function *kmeans* in Matlab® or its open-source alternative GNU Octave.



### 3.2.3 *GPS time series*

The GPS time series were created as part of the RATES project, not solely the REGINA

study. They will be made available along with the detailed descriptions in Petrie at al. (in prep

b). The time series of vertical bedrock displacement will then be accessible here: [LINK].

### 4. GRAVIMETRY DATA ANALYSIS

We investigate the Release 5 (RL05) GRACE coefficients of the Centre for Space Research

(CSR; Bettadpur, 2012) and the German Research Centre for Geosciences (GFZ; Dahle, 2013),

provided up to spherical-harmonic degree and order $j_{max}$=96 and 90 respectively in the Science

Data System (SDS). For reasons of comparison, we adopt $j_{max}$=90 for both GRACE solutions.

A temporal linear trend in the ocean bottom pressure variations modeled by the atmospheric

and oceanic background models (GAD) was re-added to the monthly solutions, according the

GRACE Science and Data System recommendation (Dobslaw et al. 2013). The GRACE

coefficients $C_{20}$ were replaced by estimates from Satellite Laser Ranging (SLR) provided by

Cheng et al. (2013). In our analysis we apply the cut-off degrees $j_{max}$=50, which has been

commonly used, as well as $j_{max} = 90$, which is considered experimental in terms of the

remaining signal content.

The determination of the rate of the gravity field change over Antarctica follows the scheme

sketched in Fig. 3. The rate of the gravity field change, expressed as equivalent water height

variations, is estimated in the spatial domain by adjusting a six-parameter function consisting

of a constant, a temporal linear trend and annual and semi-annual harmonic amplitudes. A

quadratic term was not co-estimated due to the project's focus on the rates (i.e. temporal linear

trends). It should be stated that including a quadratic term would slightly reduce the residual

uncertainties, particularly in the Amundsen Sea Sector, where an acceleration of mass balance



rates occurs that is not accounted for by interannual SMB variations of the ice sheet (see Section

4.2).

The post-processing of the GRACE coefficients follows three main steps:

*Step 1:     Optimization of de-striping filter*

Due to effects like the propagation of measurement noise and temporal aliasing, a large

proportion of the variations contained in the monthly solutions is related to noise. The noise of

the monthly solutions is lowest close to the pole and exhibits a characteristic north-south





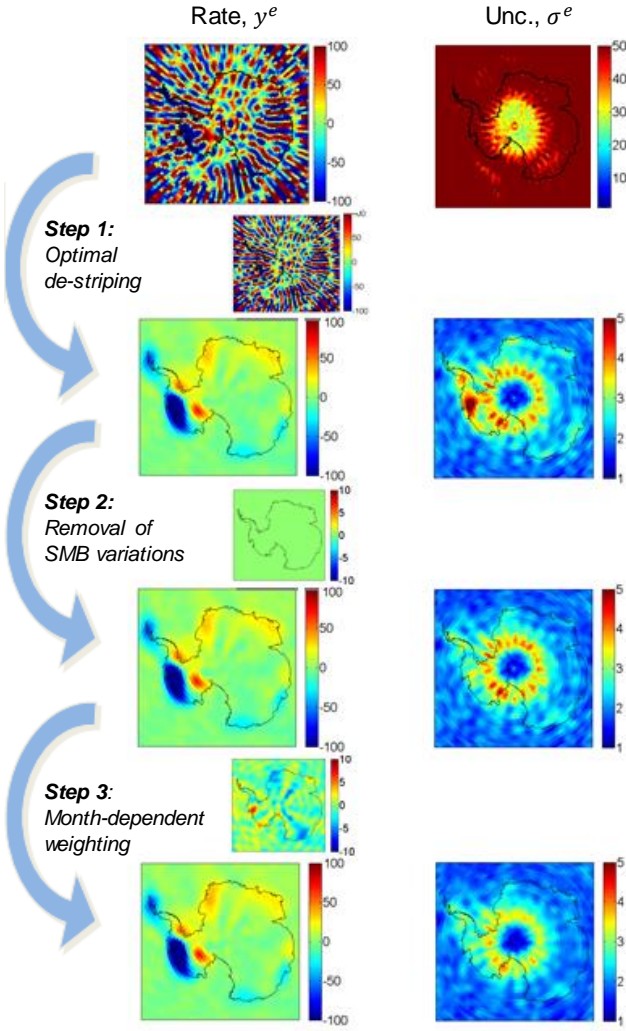

Figure 3. Post-processing steps applied to the GRACE gravity fields; shown is the impact on

the gravity field rate $y^g$ (left) and the associated RMS uncertainty $\sigma^g$ (right). Small

maps show change in the gravity field rate between two subsequent steps. Color scale

is mm w.e./yr. GRACE data is GFZ RL05a.

oriented stripe pattern. This is visible in the gravity field rate and the propagated Root-Mean-

Square (RMS) uncertainties shown in Fig. 3. In order to remove the stripe pattern, we apply the

de-correlation filter of Swenson & Wahr (2006) (hereinafter, "Swenson filter") specifically tuned to optimize the recovery of the gravity field rate over the region of Antarctica, which is detailed in Section 4.1. Fig. 3 shows that the de-striping procedure reduces the RMS uncertainty 402 of the rate by approximately one order of magnitude.

Step 2: Reduction of interannual mass variations

For isolating gravity field rates, the second step in the processing is the reduction of de- 405 trended variations of the surface mass balance, caused by accumulation events. The data set used for this purposes is the RACMO2/ANT (Lenaerts et al 2012) converted into monthly sets of spherical harmonic coefficients. The reduction of these interannual variations does not 408 change the temporal linear trend, but it reduces RMS uncertainties especially in coastal regions (Fig. 3). Details are provided in Section 4.2.

Step 3: Month-dependent weighting

The performance of the GRACE satellite system was weaker in the early mission phase 411 due to issues with the star cameras of the satellites (*C. Dahle, GFZ, pers. comm.*; Fig. 5). A rate estimate with uniform weighting of all months does not account for these variations. Therefore, 414 in the last step, month-dependent uncertainties are estimated and applied as weights during the linear regression of the temporal linear trend. This slightly changes both the resulting rate estimate, as well as its RMS uncertainties. Details are provided in Section 4.3.

Finally, after post-processing and evaluation of the gravity field rate (Section 4.4), we select the GRACE release and cut-off degree providing the lowest uncertainty level (Section 4.5) as reference input for our joint inversion for present-day ice-mass change detailed in REGINA 420 paper II (Sasgen et al. *submitted.*).



### 4.1 *Optimization of de-striping filter*

The Swenson filter has been proven to effectively reduce the typical north-south correlated

error structures of GRACE monthly solutions. The filter is based on the observation that these

structures correspond to correlated patterns in the spherical harmonic domain, namely

correlations within the coefficients of the same order and even degree, or respectively, odd

degree (Swenson & Wahr, 2006). The standard way of fitting and removing these patterns is by

adjusting polynomials to the respective sequences of spherical harmonic coefficients,

independently for individual months. Parameters to choose are the degree of the polynomial

$n_{pol}$ and the minimum order $m_{start}$ starting from which this procedure is applied. In principle,

a higher degree polynomial reduces the variability of coefficients of even / odd degree, and

results, also at lower minimum order, in stronger filtering – however, the behavior of the filter

may differ for regional applications, as discussed below. Note that tuning of other parameters

has been presented, e.g. the window width (Duan et al. 2009) or the degree range to which the

filter is applied. Chambers and Bonin (2012) have assessed these parameter options with regard

to the new GRACE RL05 solutions and global oceanic signals. Here, we perform a detailed

analysis of the choice of the Swenson filter parameters in order to optimize the signal-to-noise

characteristics of the rate of the gravity-field change over Antarctica. The resulting gravity field

rates are later used in the joint inversion for present-day ice-mass change and GIA described in

REGINA Part II.

We assess signal corruption by applying the filter to a synthetic test signal, which is based

on high-resolution elevation rates from satellite altimetry and reflects the prevailing signatures

of present-day ice-change with sufficient realism. For each choice of filter parameters, the

signal corruption is assessed as the RMS difference between the original and the filtered





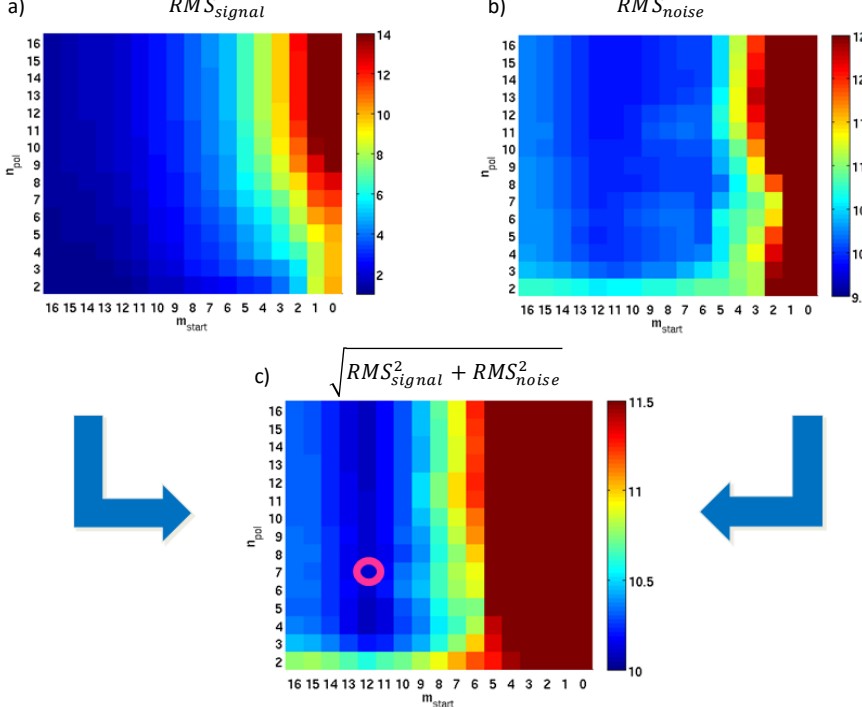

Figure 4. Effect of Swenson filter parameters $m_{start}$ and $n_{pol}$ on a) signal corruption, b) noise reduction and c) combined effect on signal and noise. RMS residuals are shown for the gravity field rates in (mm w.e./yr). The optimal choice of filter parameters $m_{start}$ = 12 and $n_{pol}$ = 7 is indicated as circle. Results are shown for GFZ RL05a with $j_{max}$ = 90.

synthetic signal, $RMS_{signal}$. The RMS is evaluated in terms of water-equivalent height per year

for the signal components within the region south of 60°S latitude.

For assessing the noise and noise reduction in the filtered fields, we face the task of

separating the noise from the geophysical signals in the gravity field rates derived from

GRACE. Here we attempt such a separation by reducing *a priori* information on the rate of ice

mass change from the GRACE fields and considering the residual as an upper bound

representation of noise. The *a priori* information is, again, based on elevation rates. For the

noise assessment we then take the RMS of the residual rates in terms of water equivalent height

per year, $RMS_{noise}$, again for the region south of 60°S latitude. Since the residual gravity field

rates may still contain some geophysical signal, we consider this noise estimate as an upper

bound for the true GRACE uncertainties. It should be stated that, after the Swenson filtering,

an additional Gaussian filtering is applied to the signal and noise models with a 200 km filter

width, which was determined to be the optimal smoothing half-width for the signal-to-noise

ratio in the GRACE spectra by Wiener optimal filtering (Sasgen et al. 2006) as reflected in the

degree-amplitude spectrum.

Fig. 4 shows the assessed signal corruption and noise reduction as a function of the two

Swenson filter parameter choices, the polynomial degree $n_{pol}$ and the minimum order $m_{start}$.

The results are shown for the gravity field expanded to degree and order $j_{max} = 90$ of the GFZ

RL05a coefficients, even though using $j_{max} = 50$ and CSR RL05 yields similar results. As

expected, the signal corruption, $RMS_{signal}$ increases with increasing strength of the Swenson

filter, that is with increasing $n_{pol}$ and the decreasing minimum order $m_{start}$. In terms of noise

reduction, we see as expected that stronger filtering (increasing $n_{pol}$; decreasing $m_{start}$)

decreases the $RMS_{noise}$ (Fig. 4), however, only for the range of filter parameters with $m_{start} \geq$

10. For $m_{start} < 10$ this pattern is reversed. A closer analysis indicates that the consideration

of the low orders into the Swenson filtering transfers energy (both from signal and noise) from

low-to-mid latitudes to the Polar Regions. This leads to a considerable signal corruption that is

only avoided by limiting the range of filter parameters in this regional analysis.

To define the optimal filter parameters a quadratic sum of the signal corruption and noise

reduction is computed, allowing us to balance both effects, the optimal values are $m_{start} =$

12 and $n_{pol} = 7$ as indicated in Fig. 4c. These filter parameters are subsequently used. For



comparison it is stated that Chambers & Bonin, 2012 find $m_{start} = 15$ and $n_{pol} = 4$ as

optimal for oceanic applications.

### 4.2 *Reduction of interannual mass variations*

Interannual variations are a major constituent of the temporal variations of the Antarctic

gravity field (Wouters et al. 2014). A large portion of the non-linear signal in geodetic mass and

volume time series is well explained by modelled SMB fluctuations (Sasgen et al. 2010;

Horwath et al. 2012). Towards the ultimate goal of isolating the linear GIA signal from time

series of mass change, we removed non-linear effects of modelled SMB variations from the

GRACE time series; for this we calculate the *monthly cumulative SMB anomalies* with respect

to the time period 1979 to 2012 obtained from RACMO2/ANT (Lenaerts et al. 2012).

We then transfer the monthly cumulative SMB anomalies in terms of their water-equivalent

height change into the spherical harmonic domain and subtract them from the monthly GRACE

coefficients. In principle, the reduction of the SMB variations from the GRACE time interval

has two effects: first, it may change the overall gravity field rate derived from GRACE,

depending on the assumption of the SMB reference period. Ideally, the reference period reflects

a state of the ice sheet in which input by SMB equals the outflow by ice discharge, and SMB

anomalies estimated for today reflect the SMB component of the mass imbalance. However,

any bias in the SMB in the reference period leads to an artificial trend in the ice sheet mass

balance attributed to SMB. This is an undesired effect, and to avoid it we de-trend the

cumulative SMB time series for the time interval coeval to the GRACE analysis (February 2003

to October 2009), before subtracting it from the gravity field rates derived from GRACE (zero

difference for *Step 2*, Fig. 3). The second effect is the reduction of the post-fit RMS residual for

this known temporal signal variation. After reducing the SMB variations, the propagated RMS



uncertainty of the derived gravity field rate becomes closer to the uncertainty level of the

GRACE monthly solutions (Fig. 3).

### 4.3 *Month-dependent weighting*

The quality of GRACE monthly solutions changes with time, for example due to changing

orbital sampling patterns (Swenson & Wahr 2006). Fig. 5 shows the temporal evolution of RMS

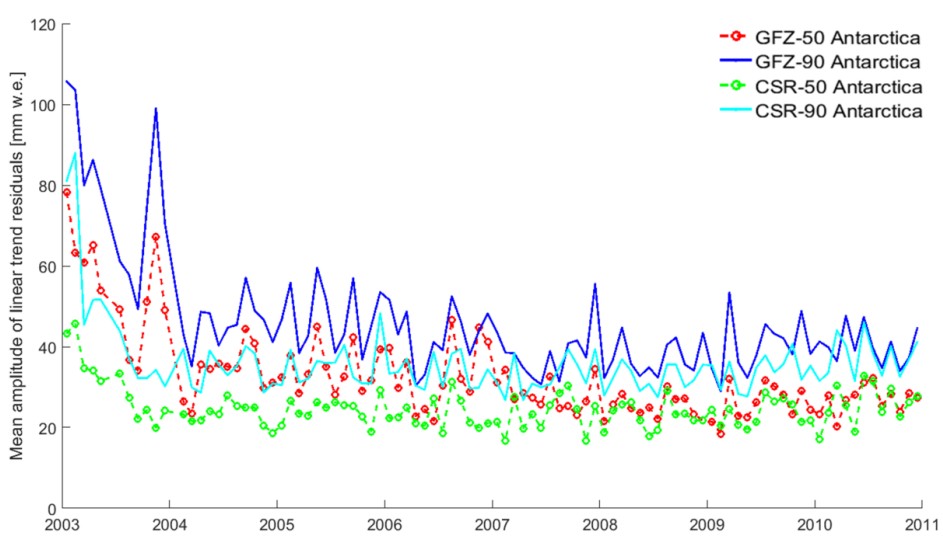

*Figure 5.  RMS uncertainty of monthly GRACE gravity fields for 2003-2011, averaged over*

*the Antarctic region south of $-60°S$ latitude. Shown are results for GFZ RL05a and*

*CSR RL05 and $j_{max} = 50$ and $j_{max} = 90$.*

uncertainties of the monthly GRACE gravity fields in the Antarctic region. Shown are residual

mass anomalies, integrated over Antarctica, after the grid-based removal of the temporal linear

trend and annual oscillation components and applying the filtering described in *Step 1*. To

improve the accuracy of the estimate of the gravity field rate, we include monthly uncertainties



as weights in our least-squares linear regression. Fig. 5 shows that these uncertainties are higher

during early 2003. Applying the monthly dependent weighting has the effect of reducing the

influence of the first months of the year 2003 on the estimated gravity field rate, which is similar

to shortening the time series, given the relatively large uncertainties. Also, the post-fit RMS

uncertainty associated with the rate reduces, if the early months of the year 2003 are excluded,

indicating that down-weighting the months from early 2003 is more beneficial than retaining a

longer time series. Altogether, the month-dependent weighting reduces the magnitude of stripe

patterns characteristic for the uncertainty of GRACE monthly solutions, and yields a more

accurate representation of propagated RMS uncertainty associated with the gravity field rates

(Fig. 3).



4.4 *Gravity field rate and uncertainty assessment*

Fig. 6 shows the estimated RMS uncertainty of the gravity field rate over Antarctica, after

post-processing. It is evident that the largest uncertainties are located in a ring south of −80°S

latitude. This is explained by the design of the Swenson filter; little or no noise reduction is

achieved close to the poles, as the gravity field is represented by near-zonal coefficients, which

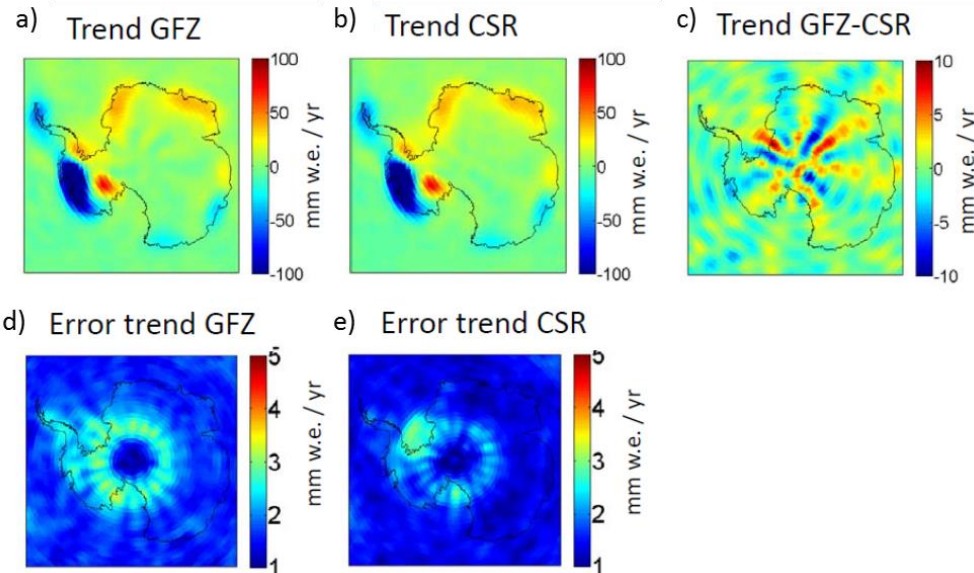

*Figure 6.  Linear trend in the GRACE gravity fields for the years 2003-2009; a) GFZ RL05a, b)*

*CSR RL05, c) difference between rates from GFZ RL05 and CSR RL05, propagated d) RMS*

*uncertainty for GFZ RL05a and e) RMS uncertainty for CSR RL05.*

pass the filter mostly unchanged ($m_{start}$ = 12). It is observed that extending the kernel of the

Swenson filter to these near-zonal coefficients ($m_{start}$ ≤ 10) creates high signal corruption

and is not suitable for the optimal rate estimate over Antarctica (see Section 4.1). Larger

uncertainties are also estimated for the Ronne and Ross ice shelf areas, which are most likely a

consequence of incomplete removal of the ocean tide signal during the GRACE de-aliasing

procedure (Dobslaw et al. 2013). It should also be stated that the RMS uncertainty estimate

does not include possible systematic errors in the GRACE solutions, e.g. due to a long-term

drift behavior of the observing system.

### 4.5 *Selection of GRACE release*

Our evaluation of the monthly GRACE uncertainties (Fig. 5), as well as the propagated

RMS uncertainty of the temporal linear trend (Fig. 6) indicates that the lowest noise level for

the Antarctic gravity field rate (February 2003 to October 2009) is currently achieved with

GRACE coefficients of CSR RL05, expanded $j_{max} = 50$. We therefore refrain from including

coefficients with $j_{max} > 50$ in order not to comprise the rate estimates by unnecessarily

increasing the noise level (see Appendix A.5, Fig. A.3). We adopt CSR RL05 with $j_{max} = 50$

as our preferred solutions for the representation of the gravity field rates over Antarctica, even

though GFZ RL05 with $j_{max} = 50$ yields very similar rates (Fig. 6). This choice is supported

by the joint inversion, as CSR RL05 with $j_{max} = 50$ provides the highest level of consistency

(lowest residual misfit) with the altimetry and GPS data sets (see REGINA Part II, Sasgen et al.

2013, Supplementary Information, Section S.3), which we interpret as minimum of spurious

signals in the trends. To account for the uncertainty related to our choice of the solution, we

consider not only RMS uncertainties of the GRACE rates but also solution differences, in the

uncertainty of the final GIA estimate (Fig. 6). The solution difference represent the absolute

deviation between trends from GFZ RL05 and CSR RL05 (February 2003 to October 2009,

cut-off degree $j_{max} = 50$). These are then summed up squared with the propagated RMS

uncertainties. It is acknowledged that the solution differences contain systematic noise arising

from the GRACE processing; the pattern and magnitude may change over time. However, they

provide a measure how much the results will change, if a GRACE release alternative to CSR



RL05 is considered. The difference between GRACE rates filtered with Gaussian smoothing of

200 km and the optimized Swenson filter together with Gaussian smoothing of 200 km is shown

in the Appendix A.5, Fig. A.4.

4.6 *Data availability*

4.6.1   *Stokes coefficients of gravity field change*

The monthly GRACE gravity field solutions from the Data System Centers GFZ and CSR

are available under ftp://podaac.jpl.nasa.gov/allData/grace/L2/ or http://isdc.gfz-potsdam.de/ as

spherical harmonic (SH) expansion coefficients of the gravitations potential (Stokes

confidents). More information is available in Bettadpur (2012). The data archive contains

temporal linear trends of the fully normalized Stokes coefficients in the 'geodetic norm'

(Heiskanen & Moritz, 1967),  complete to degree and order 90, inferred from these time series

according to Section 4,. We provide data for GFZ RL05 and CSR RL05, for the time period

2003-2009 and 2003-2013, and for various combinations of filtering. The coefficients are

organized as:

[Degree $j$], [Order $m$], [$c\_jm$], [$s\_jm$]

4.6.2   *Code for de-striping filtering*

The Matlab® function "KFF_filt" performs decorrelation filtering for sets of spherical

harmonic coefficients, typically from GRACE gravity field solutions, after the idea of Swenson

&  Wahr  (2006).  An  open-source  alternative  to  Matlab®  is  GNU  Octave

https://www.gnu.org/software/octave/.    The    function    is    called    as    KFF_filt    =

swenson_filter_2(KFF, ord_min, deg_poly, factorvec, maxdeg), where variables ord_min and

deg_poly  equal  $m_{start}$  and  $n_{pol}$, respectively, in Section 4. KFF contains the sets of spherical

harmonic coefficients in the 'triangular' format (not memory-efficient but intuitive).    For



example, for a set of coefficients with maximum degree $j_{max} = 3$ and maximum order $m_{max} =$

3, the set of coefficients is stored in a $j_{max} \times m_{max}$ matrix in the following way:

% KFF = [0    0    0    c_00 0    0    0;

%        0    0    s_11 c_10 c_11 0    0;

576    %        0    s_22 s_21 c_20 c_21 c_22 0;

%        s_33 s_32 s_31 c_30 c_31 c_32 c_33]

## 5. VISCOELASTIC MODELLING

The Earth structure of Antarctica is characterized by a strong dichotomy between east and

west, separated along the Transantarctic Mountains (e.g. Morelli & Danesi, 2004). Recent

seismic studies have produced refined maps of crustal thicknesses also showing slower upper-

mantle seismic velocities in West Antarctica, indicating a thin elastic lithosphere and reduced

mantle viscosity (An et al. 2015; Heeszel et al. 2016). Moreover, yield strength envelopes of

the Earth's crust and mantle suggest the possibility of a viscously deforming layer (DL) in the

lower part of the crustal lithosphere (Ranalli & Murphy, 1987), a few tens of km thick and with

viscosities as low as $10^{17}$ Pa s (Schotman et al., 2008). High geothermal heat flux is in

agreement with the seismic inferences of a thin elastic lithosphere and low mantle viscosity,

and would favor the presence of such a DL also in West Antarctica (Shapiro & Ritzwoller 2004;

Schroeder et al. 2014).

The choice of the viscoelastic modelling approach used to determine load-induced surface

displacements and gravitational perturbations is governed by three main requirements; i) to

accommodate a lateral variations in Earth viscosity, ii) to allow for Earth structures with thin

elastic lithosphere and low viscosity layers, in particular including a DL, and iii) to provide

viscoelastic response functions for the joint inversion of the satellite data described in REGINA

paper II (Sasgen et al. *submitted*). To meet these requirements, we adopt the time-domain

approach (Martinec 2000) for calculating viscoelastic response functions of a Maxell

continuum to the forcing exerted by normalized disc-loads of constant radius. Then, the

magnitudes and spatial distribution of the surface loads are adjusted according to the satellite

data to obtain the full GIA signal for Antarctica. The forward modelling of viscoelastic response

functions a classic topic in sold Earth modelling (e.g. Peltier & Andrews, 1976), however, their

application to inverting multiple-satellite observations for present and past ice sheet mass

changes is new and applicable to other regions, such as Greenland or Alaska.

The viscoelastic response function approach allows for high spatial resolution at low

computational cost in the numerical discretization of the Earth structure as well as in the

representation of the load and the response. In addition, we can accommodate a high temporal

resolution, which is required when considering low viscosities and associated relaxation times

of only a few decades.  The spherical harmonic cut-off degree for the simulations shown in the

following is $j_{max} = 2048$ (ca. 10 km).

### 5.1 *Load model parameters*

The load function $\sigma(t, \vartheta)$ is disc shaped with a constant radius of ca. 63 km. The radius of

63 km matches the mean radius of the discs south of 60°S of the geodesic grid (here, ICON 1.2

grid, status 2007, e.g. Wan et al., 2013), which underlie the joint inversion of the altimetry,

gravimetry and GPS observations (see REGINA paper II, Sasgen et al. *submitted.*). The

resolution of the geodesic grid is chosen to allow for an adequate representation of the load and

viscoelastic response with regard to the input data sets, while minimizing the computational

cost. The disc load experiment consists of a linear increase in the ice thickness at a rate of 0.5

Earth System
Science
Data

618    m/yr continuing until a new dynamic equilibrium state between load and response is reached.

With reference to the assumed ice density of 910 kg/m³, this thickness increase corresponds to

a mass gain of *ca.* 5.6 Gt/yr. Then, to obtain the signal component of the viscous Earth response

only, the elastic response and the direct gravitational attraction of the load are subtracted.

The experiment is designed as an *increasing* load, for example representative for the

ceasing motion of the Kamb Ice Stream (Ice Stream C; Retzlaff & Bentley, 1993), West

Antarctica. Due to linearity of the viscoelastic field equations, it is not necessary to calculate

separately the equivalent *unloading* experiment, $-\sigma(t,\vartheta)$, for example corresponding to the

past and present glacier retreat of the Amundsen Sea Sector, West Antarctica (Bentley et al.

2014 and Rignot et al. 2014, respectively). Among others, the combined inversion of the

altimetry, gravity and GPS data (REGINA paper II, Sasgen et al. *submitted.*) solves for the

magnitude and the sign of the load, allowing for ice advance as well as ice retreat.

5.2 *Earth model parameters*

We set up an ensemble of 58 simulations representing different parameterizations of the

viscosity structure (Table 5), split into West Antarctica (56 simulations) and East Antarctica (2

simulations). For West Antarctica, varied parameters are the lithosphere thickness, $h_L$ (30 to 90

km in steps of 10 km), the asthenosphere viscosity ($1 \times 10^{18}$ Pa s to $3 \times 10^{19}$ Pa s in four

steps), and the presence of a ductile lower crust, DL, with $10^{18}$ Pa s. For East Antarctica, we

employ parameter combinations appropriate for its cratonic origin with $h_L$ of 150 km and 200

km, and an asthenosphere viscosity equivalent to the upper-mantle viscosity of $5 \times 10^{20}$ Pa s.

These values lie in the range of previously applied viscosity values in Antarctica (Nield et al.

2012; Whitehouse et al., 2012; Ivins et al., 2013; van der Wal et al., 2015). For the radial

layering of the elastic properties, we adopt the Preliminary Reference Earth Model (PREM;



Dziewonski & Anderson 1981).

*Table 5. Earth model parameters associated with the disc load ensemble simulations. The viscoelastic parameterization of the Earth model is discretized in six radial layers; upper and lower crust, mantle lithosphere, asthenosphere, upper and lower mantle. The lower mantle extends down to the core mantle boundary (CMB; at the depth of 2763 km). Elastic layers are represented by a quasi-infinite viscosity of $10^{30}$ Pa s.*

| Layer | Depth (km) | Viscosity (Pa s) | Unique param. val. |
|---|---|---|---|
| **West Antarctica** | | | |
| Upper crust | 20 | $10^{30}$ | 1 |
| Lower crust DL [yes/no] | 30 | $[10^{30}/10^{18}]$ | 2 |
| Mantle lithosphere | [30, 90, steps of 10] | $10^{30}$ | 7 |
| Asthenosphere | 200 | $[1\times10^{18}, 3\times10^{18}, 1\times10^{19}, 3\times10^{19}]$ | 4 |
| Upper mantle | 670 | $5\times10^{20}$ | 1 |
| Lower mantle | CMB | $2\times10^{22}$ | 1 |
| Number of simulations West Antarctica | | | 56 |
| **East Antarctica** | | | |
| Crust | 30 | $10^{30}$ | 1 |
| Mantle lithosphere | [150, 200] | $10^{30}$ | 2 |
| Upper mantle | 670 | $5\times10^{20}$ | 1 |
| Lower mantle to CMB | CMB | $2\times10^{22}$ | 1 |
| Number of simulations East Antarctica | | | **2** |
| **Elastic earth** | | | |
| Crust and mantle to CMB | CMB | $10^{30}$ | 1 |
| **Total number of simulations** | | | **59** |

Later, in the joint inversion, the distribution of viscoelastic response functions is based on the Earth structure model of Priestley & McKenzie (2013). Priestley & McKenzie (2013) provide a global distribution of viscosity values up to a depth of 400 km, which is sampled at

the location of the geodesic grid. We then define a threshold value for the viscosity (here, 10 $^{22}$ Pas) above which the Earth response is considered purely elastic and infer the associated

thickness of the elastic lithosphere. Note that the Earth response in the equilibrium state only depends on the lithosphere thickness (independent of viscosity), which is therefore consider as the main Earth model parameters in the joint inversion. Further details are presented in REGINA paper II, Sasgen et al. *submitted.*

### 5.3 *Gravity and displacement rate response functions*

The calculated response functions for surface deformation (radial displacement) and gravity (geoid height change) are discretized along 1507 latitudinal points within the range $0 \leq$
$\vartheta \leq 90$. Simulations are typically run over 2 kyr with a temporal resolution of $\Delta t = 10$ yr (plus two time steps with constant load thickness). For East Antarctic parameterizations, the simulation period was extended to 20 kyr due to the higher upper-mantle viscosities and
associated slower relaxation. However, note that the ratio of geoid-height change versus radial displacement falls off to 1/e after ca. 2 kyr of simulation (Appendix A.6, Fig. A.5). The forcing expected in central East Antarctica is an increase in accumulation towards present-day
conditions after ca. 7 ka BP (van Ommen et al. 2004), justifying also the use of equilibrium kernels for East Antarctica. The time derivatives of the radial displacement $\dot{u}$ and of the geoid height change $\dot{e}$ are calculated with a central difference scheme.




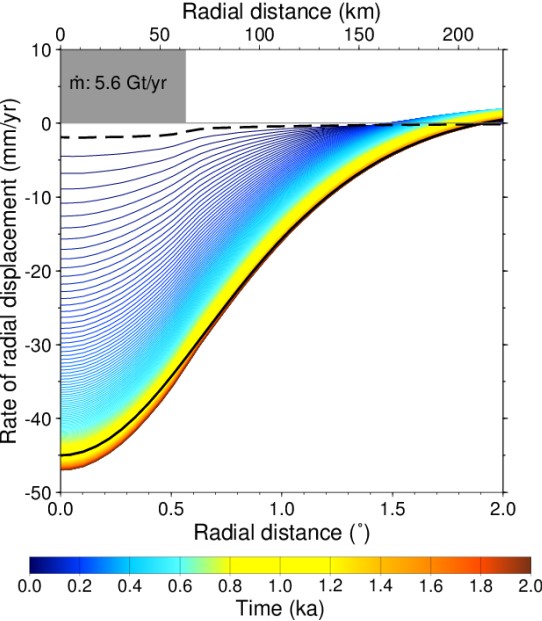

*Figure 7. Displacement rates over the simulation period of 2 kyrs, for an exemplary set of Earth model parameters ($h_L = 30\ km$; $\eta_{AS} = 1 \times 10^{18}\ Pa\ s$). Shown is the load dimension (grey shading), as well as the instantaneous elastic response (dashed black line) and viscoelastic relaxation only after 2 kyr and no load change (solid black line). The other curves show the rates for the time epoch indicated by the color scale.*

Examples of response functions to the loading detailed in Section 5.1 for the rate of radial

displacement, $\dot{u}$, and rate of geoid-height change, $\dot{e}$, are shown in Figs 7 and 8, respectively.

Instantaneously, the increasing load, $\dot{\sigma}(t) = \mathrm{const.}$, induces an elastic response that is

characterized by subsidence and an increase in the direct gravitational potential (dashed lines

in Fig. 7 and Fig. 8, respectively). This is the elastic response function adopted in the joint

inversion. Note that the elastic response function will not differ between East and West

Antarctica, as it is entirely based on the distribution of densities and elastic parameters provided

by the PREM. As the load build-up continues, the instantaneous response is followed by the



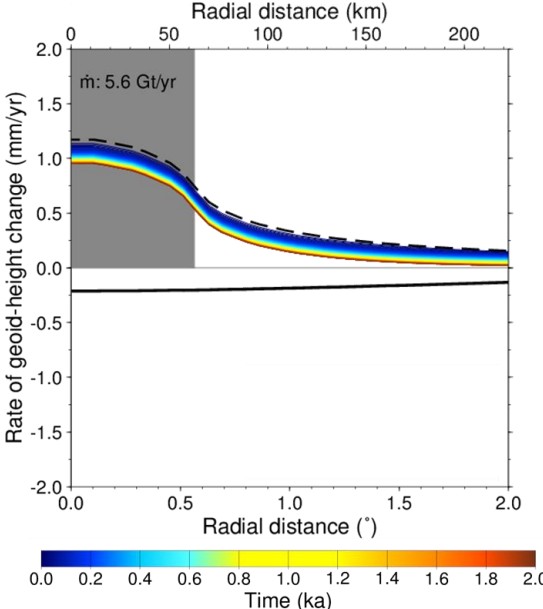

Figure 8. *Same as Fig. 7, but for the rate of geoid-height change and Earth model parameters*

$h_L = 90\ km$ ; $\eta_{AS} = 1 \times 10^{18}\ Pa\ s$. *Note the change in sign in the rate when the increase*

*in direct gravitational attraction through load increase ceases after 2 kyr (black solid line*

*vs. colored lines).*

viscoelastic response, which depends in timing and magnitude on the underlying lithosphere

and viscosity structure, further increasing the displacement rates, $\dot{u}$ (blue to red lines in Fig. 7).

The compensation by solid Earth deformation is reflected in the decreasing geoid rate, $\dot{e}$ (Fig.

8). After a certain time, which depends on the value of the asthenosphere viscosity, a new

dynamic equilibrium state is reached at which $\dot{u}$ and $\dot{e}$ do not change in time any more. In the

last two time steps, the load is kept constant ($\dot{\sigma}(t) = 0$), and the responses in $\dot{u}$ and $\dot{e}$ are only

caused by the relaxation of the Earth's viscoelastic deformation (solid black line in Figs. 7 and

8), which is the viscoelastic response function adopted in the joint inversion.



### 5.4 *Discussion of effects of selected earth model parameterizations on GIA response*

Fig. 9 shows the response of $\dot{u}$ for four end-member sets of Earth model parameters with

thick lithosphere, weak asthenosphere $(TkWk: h_L = 90\ km\ ;\ \eta_{AS} = 1 \times 10^{18} Pa\ s)$, thick

lithosphere, strong asthenosphere $(TkSg: h_L = 90\ km\ ;\ \eta_{AS} = 3 \times 10^{19} Pa\ s)$, thin

lithosphere, weak asthenosphere $(TnWk: h_L = 30\ km\ ;\ \eta_{AS} = 1 \times 10^{18} Pa\ s)$ and thick

lithosphere, strong asthenosphere $(TnSg: h_L = 30\ km\ ;\ \eta_{AS} = 3 \times 10^{19} Pa\ s)$, without a

ductile layer, DL. In this context, thick / thin and strong / weak refer to values in comparison to

the 'average' value of the ensemble for West Antarctica; an elastic lithosphere of thickness 90

687    km (here, '*Tk*') is in the range of global average continental lithosphere usually applied in GIA

studies (e.g. Peltier, 2004), or that of East Antarctica (150 to 200 km). Fig. 10 shows the

response in $\dot{u}$ for the same end-member set of Earth model parameters with a DL included. It

should be stated that the Earth structure with $h_L = 30$ km *and* a DL is considered very extreme,

because in this case the ductile layer extends down to the asthenosphere and an elastic mantle

lithosphere is missing.

Fig. 9 and 10 show that for the weak asthenosphere $(\eta_{AS} = 1 \times 10^{18}$ Pa s), viscoelastic

deformation is visible already after one decade of loading (or unloading), leading to

considerably larger subsidence rates compared to the purely elastic case even on very short time

scales. For these Earth model parameters, a new dynamic equilibrium state is achieved within

a few centuries. The rates of subsidence in this equilibrium then primarily depend on the support

provided by the flexure of the elastic lithosphere.

For the extreme *TnWk* case, equilibrium rates of $-45$ mm/yr are achieved at the load

centre, and considerable subsidence of $-20$ mm/yr already occurs after ten years of loading

(Fig. 9). Increase in asthenosphere viscosity (*TnSg* case) reduces the viscous material transport



and leads to a slower adjustment towards the dynamic equilibrium state, which takes more than

1 kyr. It should be stated that in our definition of the ensemble parameters, reducing the

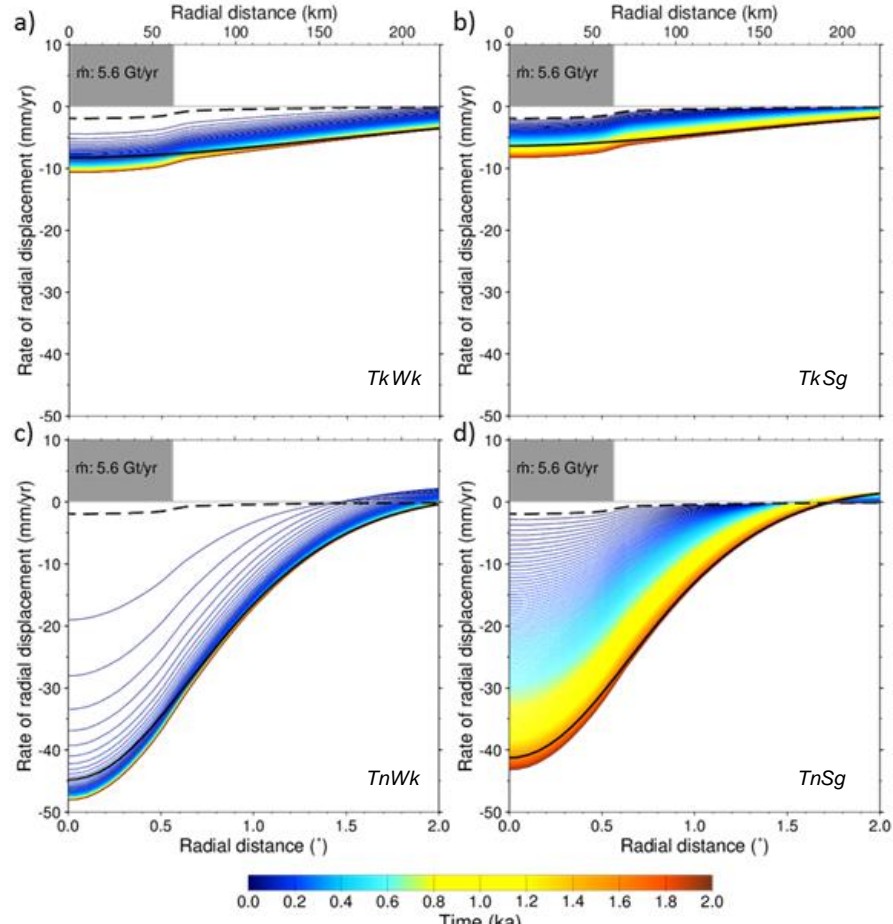

*Figure 9. Same as Figure 7, but for four end-member sets of Earth model parameters, without a*

*DL and lithosphere thickness / asthenosphere viscosity of a) $h_L = 90\ km$ / $\eta_{AS} =$*

*$1 \times 10^{18} Pa\ s$ (TkWk), b) $h_L = 90\ km$ / $\eta_{AS} = 3 \times 10^{19} Pa\ s$ (TkSg), c) $h_L = 30\ km$ /*

*$\eta_{AS} = 1 \times 10^{18} Pa\ s$ (TnWk) and d) $h_L = 30\ km$ / $\eta_{AS} = 3 \times 10^{19} Pa\ s$ (TnWk).*



lithosphere thickness in turn increases the thickness of the asthenosphere (bottom depth of

asthenosphere is fixed), which facilitates lateral material transport inside the asthenosphere.

The consideration of the DL in the Earth structure causes a thinning of the effective elastic

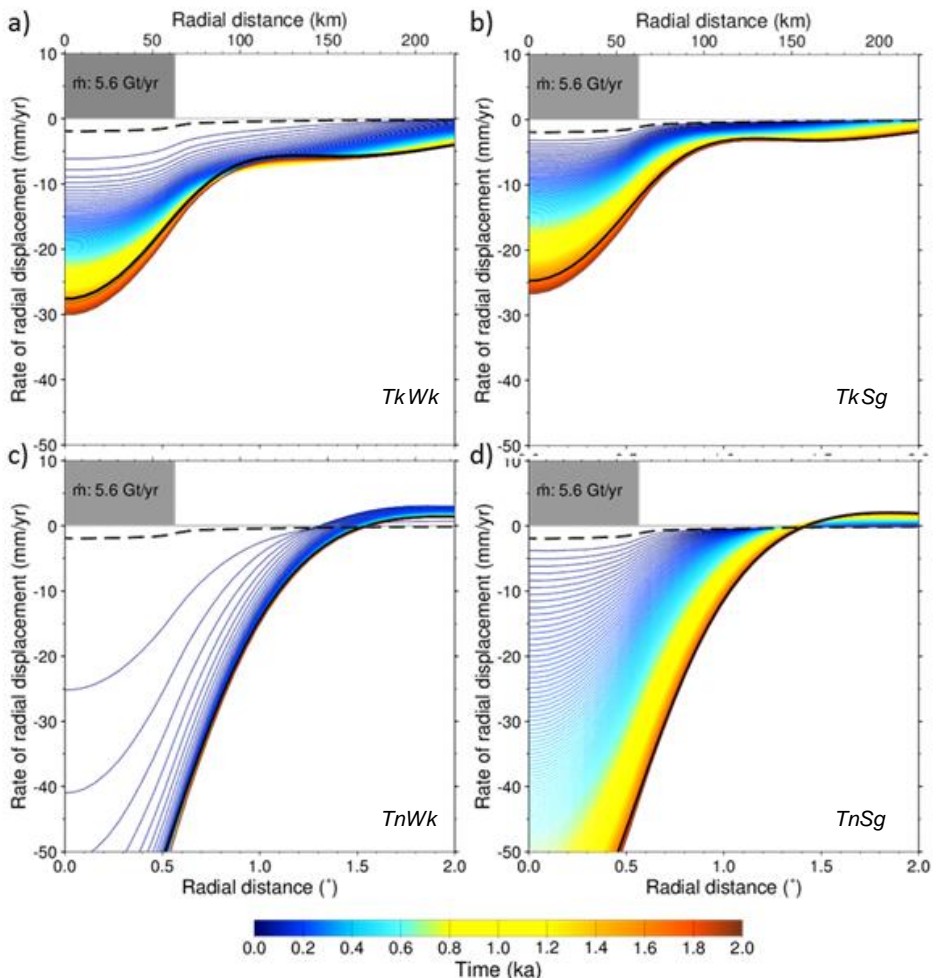

Figure 10. Same as Figure 9, a) TkWk, b) TkSg (b), TnWk (c) and TnSg (d), but with the Earth structure including a DL.

lithosphere. As a consequence, greater and more localized subsidence rates are produced for all

sets of parameters (Fig. 10). Interestingly, in case of a thick elastic lithosphere (90 km), the

radial displacement exhibits a local minimum at around 120 and 160 km distance from the load

centre (Fig. 10), which is a consequence of the viscous material transport inside the ductile

layer. The maximum equilibrium rate of $-76$ mm/yr is achieved for the *ThWk* case with DL,

where the viscous deformation leads to rates of $-25$ mm/yr already after 10 yrs of loading.

### 5.5 *Assumptions and limitations*

Although the approach of modelling response functions to axisymmetric disc loads and

subsequently superposing them is very efficient in terms of the computational cost, this

simplification introduces some limitations. First, the superposition of response functions

representing different Earth structures neglects the transmission of stresses between these

regions — a problem that can only be resolved with fully three-dimensional solid Earth

modelling (e.g. van der Wal et al., 2015). The largest impact for the displacement rates is

expected in regions with lateral contrasts in lithosphere thickness and mantle viscosity such as

the Transantarctic Mountains. Second, the constant disc radius of about 63 km implies that

finer-scale deformation cannot be resolved. Although this resolution is adequate for interpreting

GRACE data (spatial half-wavelength of *ca.* 200 km) smaller-scale loading excitement may be

necessary for interpreting local GPS measurements near to the loading, particularly for the

elastic response to present-day glacial changes.

### 5.6 *Data availability*

#### 5.6.1 *Viscoelastic kernels*

Output files contain 1507 latitudinal points ($0 \leq \vartheta \leq 90$) covering a region greater than the

size of the Antarctic domain, as well 203 time steps of West Antarctica (213 time steps for East

Antarctica, because of extending the simulation period to 15 kyrs). The time derivative of the





radial displacement, $u$, is calculated with a central difference scheme, $y^u := $ [u(t+Δt/2)-u(t-

Δt/2)]/Δt. The difference between two time steps is Δt=10 yr. The same applies to the rate of

geoid-height change, $y^g$. Note that the load is constant during the last two time steps (no rate

of change); therefore, the kernels represent the viscoelastic relaxations only, without the

instantaneous elastic deformation or the direct gravitational attraction of the load.

The results are stored independently for the rheology of East and West Antarctica, the latter

with and without a ductile layer in the elastic part of the lithosphere. The data are stored in a

Matlab®    file    format,    which    is    also    readable    with    GNU    Octave

https://www.gnu.org/software/octave/ .

- 'Viscoel_response_WA_with_DL.mat' – Response functions for West Antarctica

*with* ductile layer

- 'Viscoel_response_WA_no_DL.mat' – Response functions for West Antarctica

*without* ductile layer

- 'Viscoel_response_EA_no_DL.mat' – Response functions for East Antarctica

*without* ductile layer

- 'Time_EA_rheo.mat / Time_WA_rheo.mat' – Time [kyr] file related to response

file for East and West Antarctica

- 'Coord_Co-Latitude.mat' – Co-latitude [°] of the response functions

The response matrix summary the data as follows:

*West Antarctica:*

VE_WA_no_DL has the following entries, [HL, AV, LAT, TIME, VAR]

HL: Lithosphere thickness; 30 km, 40 km, …, 90 km (7 entries)

AV :  Asthenosphere  Viscosity;   $1 \times 10^{18} \, Pa \, s$,  $3 \times 10^{18} \, Pa \, s$,  $10 \times 10^{18} \, Pa \, s$,  $30 \times$



$10^{18}\ Pa\ s$ (4 entries)

LAT: Latitude grid node, corresponding to file 'Coord_Co-Latitude.mat' (1537 entries)

TIME: Time, corresponding to file 'Time_WA_rheo.mat' (202 entries)

VAR: Variable type; 1: rate of radial displacement in mm/yr, 2: rate of geoid-height change in

mm/yr.

The response kernels for *East Antarctica* are organized analogue, [HL, LAT, TIME, VAR]

HL: Lithosphere thickness; 150 km, 200 km (2 entries)

Note that the asthenosphere and upper mantle viscosity is constant at $5 \times 10^{20}\ Pa\ s$ and

therefore has no entry.

The spectral resolution underlying these fields is spherical-harmonic cut-off degree 2048.

The user should apply an adequate smoothing filter when using for inverting GRACE gravity

fields. Filtered kernels are available upon request by the author.

### 5.6.2 *Geodesic grid*

The computation of the geodesic grid is not an original contribution of the authors, but

based on the grid generator of the ICON GCM project, http://icon-downloads.zmaw.de/. For

completeness, we provide the data set with disc locations based. An alternative resource for

downloading geodesic grids at different resolutions in netCDF format can be found here:

http://kiwi.atmos.colostate.edu/BUGS/geodesic/ .

The files format is:

vert-7.mask.cont_and_shelf.re.dat: Longitude [°], Latitude [°]

vert-7.mask.cont_and_shelf.re.proj.dat: X [km], Y [km], (projected coordinates, WGS-84,

Polar Stereographic, 71°S true latitude, 0°E central longitude)



### 5.6.3 *Lithosphere thickness*

The thickness of the elastic lithosphere at the locations of the geodesic grid for different

values of the viscosity threshold applied to the data set of Priestley & McKenzie, 2013.

lith_thresh_21.disc.txt (threshold $10^{21}$ Pa s, thicker lithosphere)

lith_thresh_22.disc.txt (threshold $10^{22}$ Pa s, lithosphere adopted in the GIA estimate)

lith_thresh_23.disc.txt (threshold $10^{23}$ Pa s, thinner lithosphere)

The 1175 entries correspond to the locations of the geodesic grid (Section 5.6.2).

### 5.6.4 *Open source code for viscoelastic modelling*

The opens source software package SELEN allows the computation of the Maxwell-

viscoelastic Earth response to user-defined ice sheet evolutions, in particular also a simplified

disc-load forcing as presented in this paper. The program is downloadable at:

https://geodynamics.org/cig/software/selen/

### 6. CONCLUSIONS

In this paper, we have presented refined temporal linear trends of surface elevation, gravity

field change and bedrock displacement based on Envisat/ICESat (2003-2009), GRACE (2003-

2009) and GPS (1995-2013.7), respectively. In addition, we have performed forward modelling

of the viscoelastic response of the solid Earth to a disc-load forcing. These response functions

are particularly suited to represent the distinct geological regimes of East and West Antarctica

in the joint inversion of multiple satellite data. Similarly, the functions can be applied to the

other geographical regions as well. The data and code necessary to reproduce our results, or

apply our approach to a different problem, is provide at www.pangea.de,

https://doi.pangaea.de/10.1594/PANGAEA.875745.



We have refined surface-elevation rates for the Antarctic ice sheet for the time interval

2003-2009 by combining Envisat and ICESat altimetry data. The straightforward compositing

approach performs a grid-based comparison of the noise in the elevation rates obtained from

Envisat and ICESat. For large parts of the ice sheet, the elevation rate is based on ICESat data,

particularly, along for the rough terrain along coast, as well as close to the Pole (polar gap of

Envisat). Envisat contributes in some low-relief areas in East Antarctica and along the Antarctic

Peninsula, as well as along single spurious ICESat tracks. Thus, the composite elevation rates

are maximized in terms of spatial coverage and minimized in term of uncertainties.

The GPS processing carried out as part of the RATES and REGINA projects has produced

a comprehensive data set of Antarctic 118 GPS records, which, for continuous sites,

spans a longer time interval (1995-2013) than those of previous studies (Thomas et al.

(2011), 1995-2011; Argus et al.  (2014), 1994-2012; Martín-Español et al.  (2016b), 2009-

2014). The ensemble processing done for the REGINA project has allowed us to assess the

contribution of systematic error sources. In addition, for sites where there is potential doubt

over the quality of the metadata or the behaviour of the site, we have adopted a 'conservative

but realistic' approach to assigning new confidence limits. The screening of GPS data for

outliers involved careful manual assessment, encompassing the review of measurement logs

and notes on problems in the field. The data quality is reflected in the uncertainty estimates for

the GPS rates, which therefore represents more reliable input data than GPS rates based on

processing without manual intervention.

We have optimized the post-processing sequence for estimating the temporal linear trend

and its uncertainty in the GRACE gravity field solutions for the region of Antarctica. In

particular, we have derived optimal parameters for de-striping the monthly gravity fields over

Antarctica according to Swenson & Wahr (2006). In addition, we have removed de-trended interannual SMB fluctuation from the GRACE time series, to obtain a more representative

uncertainty estimate based on the post-fit RMS residual. We have included month-dependent weighting in the least-squares estimate of the gravity field rates to account for the varying quality of the monthly GRACE solutions. The optimization of the de-correlation filter of

Swenson & Wahr (2006) to the signals expected in Antarctica reduced the residual uncertainty and improved the reliability of inferred mass anomalies.

With the aim of joining the multiple satellite data using the knowledge of the geophysical

processes involved, we have calculated elastic and viscoelastic response functions of the solid Earth. The viscoelastic response functions represent the gravity field change and surface displacement to a disc-load forcing for a variety of Earth model parameters; particularly,

however, values of mantle viscosity and lithosphere thickness strongly varying between the distinct geological regimes of West and East Antarctica.

In particular, we have investigated the effect of a ductile layer in the crustal lithosphere on

the viscoelastic rebound signature. We show that for moderate load changes of 0.45 m/yr water-equivalent (here, applied as disc load with a radius of ca. 63 km), uplift rates reach the cm/yr level within decades assuming asthenosphere viscosities $< 10^{19}$ Pa s and lithosphere thickness

$< 50$ km; both plausible values for parts of West Antarctica. Including a ductile layer in the crustal lithosphere further attenuates the uplift rates and localizes the deformational response. This suggests that GIA in West Antarctica may locally be a result of more recent, centennial

load changes, most notably in the Amundsen Sea Embayment and in part of the Antarctic Peninsula (Nield et al. 2012). Similar conclusion were reached by Ivins & James (2005) and Nield et al. (2014).

The advantage of the viscoelastic response kernels is that a meaningful ratio of gravity disturbance versus surface displacement is calculated for each choice of the Earth model parameters, avoiding the approximation with an average rock density (e.g. Riva et al. 2009;

Gunter et al. 2014). Using the response functions allows us to reconcile GIA signatures with measurements of large bedrock uplift and small gravity field increase in the Amundsen Sea Embayment, associated with weak Earth structures. Clearly, the response functions adopted

here represent only the viscoelastic equilibrium state and, thus, are considered only an intermediate step to full dynamic modelling of the GIA response. Nevertheless, this approximation represents a significant improvement of other joint inversion methods, as it bases

the joint inversion on physically meaningful response kernels. With extra data on the past ice evolution, such as Paleo thickness rates, our approach can be expanded to address the temporal evolution as well.

In the succeeding paper REGINA part II (Sasgen et al. *submitted*), we perform the joint inversion for present-day ice-mass changes and GIA in Antarctica, based on the input data sets and viscoelastic response functions presented here. We validate our results using forward-

modelling results and other empirical models, and show the impact on CryoSat-2 volume and GRACE mass balances, respectively. Note, however, that the post-processing methods and viscoelastic functions presented here are applicable also to other geographical regions with

superimposed present-day mass change and GIA signatures.

## 7. DATA AVAILABILITY

    The altimetry, gravimetry, GPS and viscoelastic modelling data used in this project are

available at https://doi.pangaea.de/10.1594/PANGAEA.875745 in the www.pangea.de archive.

The data description and user documentation are given for each data type within the respective

subsection of this paper (Sections 2 to 5).

**AUTHOR CONTRIBUTION**

Ingo Sasgen conceived, managed and summarized this study with support of Mark R. Drinkwater. Alba Martín-Español, Bert Wouters and Jonathan L. Bamber performed the

873 altimetry analysis. Alexander Horvath, Martin Horwath and Roland Pail undertook the gravity field analysis, Elizabeth J. Petrie and Peter J. Clarke analysis and clustered the GPS data with critical input from Terry Wilson. Volker Klemann and Hannes Konrad performed the

876 viscoelastic modelling, with contributions from Ingo Sasgen. All authors were involved in writing and reviewing this manuscript.

**COMPETING INTEREST**

The authors declare that they have no conflict of interest.

**ACKNOWLEDGEMENTS**

The www.regina-science.eu work was enabled through CryoSat+ Cryosphere study

funding from the Support To Science Element (STSE) of the European Space Agency (ESA) Earth Observation Envelope Programme. I.S. acknowledges additional funding through the German Academic Exchange Services (DAAD) and DFG grant SA1734/4-1 and P.J.C. and

E.J.P. from UK NERC grant NE/I027401/1 (RATES project). We thank Thomas Flament and Frederique Rémy for the Envisat data and Veit Helm for providing the AWI L2 CryoSat-2 re-tracked and corrected elevation measurements. The GPS data used was mainly downloaded

from publically available archives. We acknowledge work done by the International GNSS Service (Dow et al., 2009), UNAVCO and the Scientific Committee on Antarctic Research in maintaining such archives, together with the efforts of all the GPS site operators in collecting



891 and making available the data, a particularly challenging task in Antarctica (see Table S2 for

more information on individual sources).





## APPENDIX

*A.1 ICESat campaigns and operation periods*

*Table A.1. ICESat 633 Level 2 data for the time span February 2003 until October 2009 used in this study.*

| Start Date | End Date | Days in Operation | Laser Identifier |
|---|---|---|---|
| 20/02/2003 | 29/03/2003 | 38 | 1AB |
| 25/09/2003 | 19/11/2003 | 55 | 2A |
| 17/02/2004 | 21/03/2004 | 34 | 2B |
| 18/05/2004 | 21/06/2004 | 35 | 2C |
| 03/10/2004 | 08/11/2004 | 37 | 3A |
| 17/02/2005 | 24/03/2005 | 36 | 3B |
| 20/05/2005 | 23/06/2005 | 35 | 3C |
| 21/10/2005 | 24/11/2005 | 35 | 3D |
| 22/02/2006 | 28/03/2006 | 34 | 3E |
| 24/05/2006 | 26/06/2006 | 33 | 3F |
| 25/10/2006 | 27/11/2006 | 34 | 3G |
| 12/03/2007 | 14/04/2007 | 34 | 3H |
| 02/10/2007 | 05/11/2007 | 37 | 3I |
| 17/02/2008 | 21/03/2008 | 34 | 3J |
| 04/10/2008 | 19/10/2008 | 16 | 3K |
| 25/11/2008 | 17/12/2008 | 23 | 2D |
| 09/03/2009 | 11/04/2009 | 34 | 2E |
| 30/09/2009 | 11/10/2009 | 12 | 2F |



### A.2 Firn compaction and SMB corrections

We apply rates of firn compaction, $h_{comp}$, using output of the firn compaction model

provided by Ligtenberg (2011), which is driven by RACMO2/ANT (Lenaerts 2010). However,

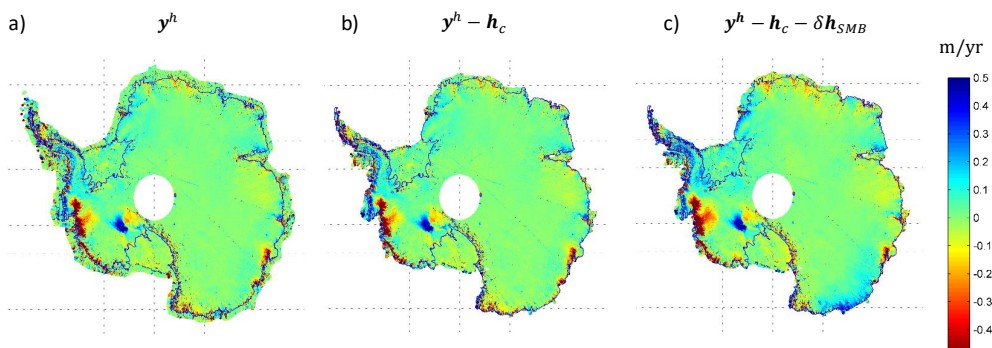

*Figure A.1 Rate of elevation change $y^h$ (m/yr), derived from a) ICESat/Envisat initial data, b)*

*ICESat/Envisat minus firn compaction $h_{comp}$, and c) ICESat/Envisat minus firn compaction*

*$h_{comp}$ and modelled SMB anomalies $\delta h_{SMB}$.*

we do not apply a correction for anomalies in the surface-mass balance (SMB), $\delta h_{SMB}$, as e.g.

undertaken by Gunter et al. (2014), due to the problem of defining an adequate reference period

for the ice sheet. The impact of each correction is shown in Fig. A.1.



*A.3 Flowchart of estimation process for Antarctic GPS site time series.*

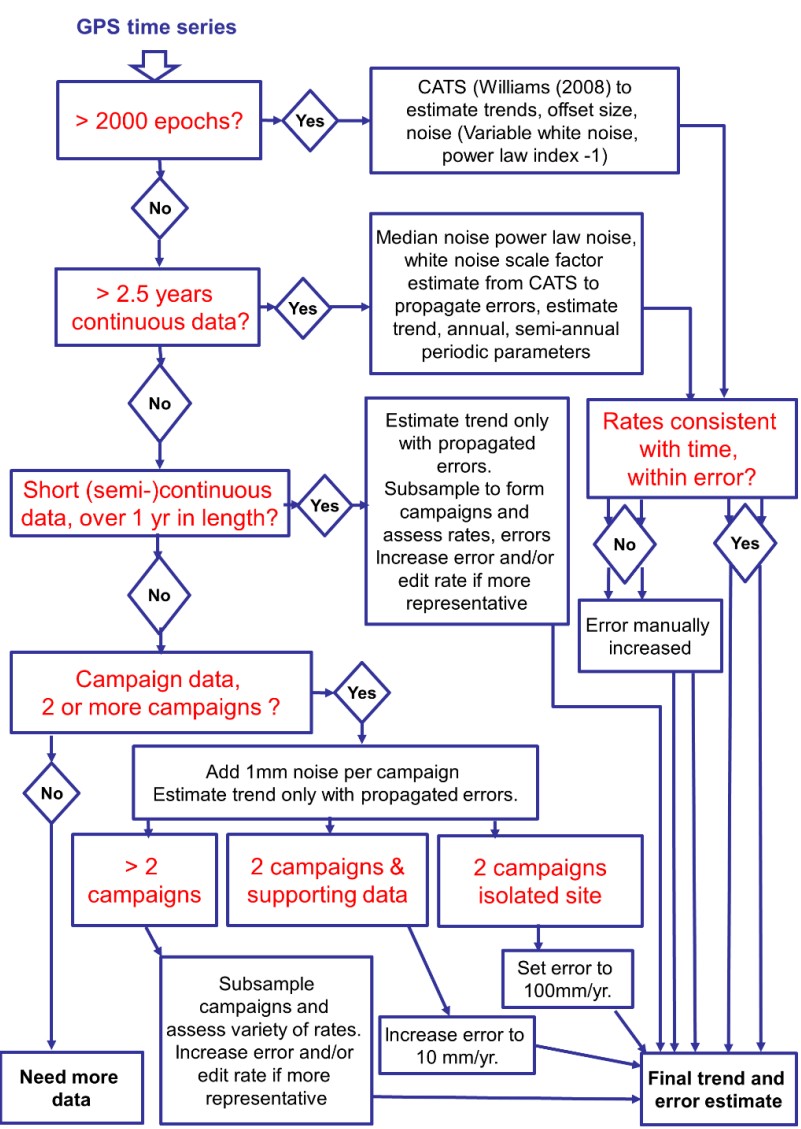

*Figure A.2 Flowchart showing the estimation process for the temporal linear trends of the*

*bedrock for Antarctic GPS site timeseries. After Petrie et al. (2016) (SCAR poster)*





*A.4 Uplift rates at all GPS site used in this study*

*Table A.2 GPS uplift rates for this study. The columns are: site name, estimated uplift rate $y^u$ (mm/yr), estimated uncertainty $\sigma^u$ (mm/yr), rate method, uncertainty method, approx latitude (dec. degrees), approx. longitude (dec. degrees). Methods are: cats: estimated by the CATS noise analysis software ('cats'), median uncertainty from CATS sites propagated ('prop'), manual intervention in rate due to potential systematic uncertainties ('rman') and manual intervention in uncertainties due to potential systematic errors ('eman').*

| Site name | $y^u$ | $\sigma^u$ | Method $y^u$ | Method $\sigma^u$ | Lat. (°) | Lon. (°) | Doi/ data source or description |
|---|---|---|---|---|---|---|---|
| aboa | 0.6 | 0.5 | cats | cats | -73.04 | -13.41 | Finnish Geodetic Institute |
| brip | 1.4 | 0.7 | cats | cats | -75.80 | 158.47 | doi:10.7283/T5W09473 |
| buri | 2.3 | 0.7 | cats | cats | -79.15 | 155.89 | doi:10.7283/T5RB72W7 |
| cas1 | 1.5 | 0.2 | cats | cats | -66.28 | 110.52 | IGS: Dow et al. (2009) |
| cote | 1.4 | 0.7 | cats | cats | -77.81 | 162.00 | doi:10.7283/T5GT5KGN |
| crar | 0.7 | 0.4 | cats | cats | -77.85 | 166.67 | UNAVCO* |
| dav1 | -1.6 | 0.6 | rman | eman | -68.58 | 77.97 | IGS: Dow et al. (2009) |
| dum1 | -0.3 | 0.3 | cats | cats | -66.67 | 140.00 | IGS: Dow et al. (2009) |
| flm5 | 2.0 | 0.6 | cats | cats | -77.53 | 160.27 | doi:10.7283/T5V40SH6 |
| ftp4 | 1.9 | 0.6 | cats | cats | -78.93 | 162.56 | doi:10.7283/T5B27SKD |
| maw1 | -0.4 | 0.2 | cats | cats | -67.60 | 62.87 | IGS: Dow et al. (2009) |
| mcm4 | 0.8 | 0.2 | cats | cats | -77.84 | 166.67 | IGS: Dow et al. (2009) |
| min0 | 2.0 | 0.8 | cats | cats | -78.65 | 167.16 | doi:10.7283/T5TM78BX |
| ohi2 | 3.4 | 2.0 | cats | eman | -63.32 | -57.90 | IGS: Dow et al. (2009) |
| palm | 4.8 | 3.0 | cats | eman | -64.78 | -64.05 | IGS: Dow et al. (2009) |
| ramg | 2.4 | 0.8 | cats | cats | -84.34 | 178.05 | doi:10.7283/T51N7ZFR |
| rob4 | 1.1 | 0.5 | cats | cats | -77.03 | 163.19 | doi:10.7283/T5NC5ZG8 |
| sctb | 0.9 | 0.5 | cats | cats | -77.85 | 166.76 | doi:10.7283/T5CF9N6P |
| syog | 1.1 | 0.2 | cats | cats | -69.01 | 39.58 | IGS: Dow et al. (2009) |
| tnb1 | 0.1 | 0.5 | cats | cats | -74.70 | 164.10 | Dubbini et al. (2010) |
| vesl | 0.4 | 0.3 | cats | cats | -71.67 | -2.84 | IGS: Dow et al. (2009) |
| a351 | -0.9 | 1.8 | prop | eman | -72.91 | 74.91 | Geoscience Australia** |
| a368 | -0.2 | 1.2 | prop | eman | -74.29 | 66.79 | Geoscience Australia** |
| arct | -0.1 | 4.4 | prop | eman | -80.04 | -80.56 | SCARP*** |
| art1 | -3.1 | 10.0 | prop | eman | -62.18 | -58.90 | Dietrich et al. (2004) |
| back | 16.8 | 5.0 | prop | eman | -74.43 | -102.48 | doi:10.7283/T5D21VWM |
| bean | 2.1 | 4.3 | rman | eman | -75.96 | -69.30 | doi:10.7283/T55Q4T6R |



| | | | | | | |
|---|---|---|---|---|---|---|
| belg | **-1.4** | 0.7 | prop | prop | -77.87 | -34.63 | Dietrich et al. (2004) |
| benn | **9.3** | 1.9 | prop | prop | -84.79 | -116.46 | doi:10.7283/T5891447 |
| berp | **25.2** | 0.7 | prop | prop | -74.55 | -111.88 | doi:10.7283/T54J0CC2 |
| bhil | **2.9** | 4.4 | rman | eman | -66.25 | 100.60 | Geoscience Australia** |
| bren | **3.1** | 1.1 | rman | eman | -72.67 | -63.03 | doi:10.7283/T52V2D7X |
| capf | **4.0** | 1.4 | rman | eman | -66.01 | -60.56 | doi:10.7283/T5XP736P |
| cjam | **-2.3** | 100.0 | prop | eman | -63.10 | -62.72 | SCARP*** |
| clrk | **3.6** | 1.4 | prop | prop | -77.34 | -141.87 | doi:10.7283/T5MK6B6C |
| coat | **-0.1** | 7.3 | prop | eman | -77.81 | 162.00 | Raymond et al. (2004) |
| crdi | **2.1** | 0.6 | prop | prop | -82.86 | -53.20 | doi:10.7283/T5C24TQS |
| cwal | **0.4** | 100.0 | prop | eman | -63.25 | -62.18 | SCARP*** |
| dal1 | **4.9** | 34.4 | prop | eman | -62.24 | -58.68 | Dietrich et al. (2004) |
| dall | **-17.0** | 100.0 | prop | eman | -62.24 | -58.66 | Dietrich et al. (2004) |
| devi | **1.9** | 1.0 | prop | prop | -81.48 | 161.98 | doi:10.7283/T57942Z0 |
| dupt | **11.5** | 1.1 | prop | prop | -64.81 | -62.82 | doi:10.7283/T5KD1W62 |
| eacf | **-4.8** | 15.0 | rman | eman | -62.08 | -58.39 | Brazil |
| elph | **6.3** | 100.0 | prop | eman | -61.22 | -55.14 | SCARP*** |
| esp1 | **5.6** | 100.0 | prop | eman | -63.40 | -57.00 | Dietrich et al. (2004) |
| fall | **4.8** | 1.3 | prop | prop | -85.31 | -143.63 | doi:10.7283/T53J3B84 |
| ferr | **-5.5** | 31.0 | rman | eman | -62.09 | -58.39 | Dietrich et al. (2004) |
| fie0 | **-0.9** | 1.9 | prop | prop | -76.14 | 168.42 | doi:10.7283/T5KK993F |
| flm2 | **3.8** | 11.7 | rman | eman | -77.53 | 160.27 | doi:10.7283/T53T9FHJ |
| fonp | **13.5** | 1.8 | prop | prop | -65.25 | -61.65 | doi:10.7283/T5668BG6 |
| for1 | **-0.2** | 2.9 | prop | eman | -70.78 | 11.83 | Dietrich et al. (2004) |
| for2 | **-0.3** | 2.7 | prop | eman | -70.77 | 11.84 | Dietrich et al. (2004) |
| fos1 | **3.1** | 1.3 | prop | eman | -71.31 | -68.32 | doi:10.7283/T54T6GF7 |
| frei | **-4.4** | 0.7 | prop | prop | -62.19 | -58.98 | Bevis et al. (2009) |
| ftp1 | **-2.2** | 3.4 | prop | eman | -78.93 | 162.56 | doi:10.7283/T53T9FHJ |
| gmez | **1.5** | 4.8 | rman | eman | -73.89 | -68.54 | doi:10.7283/T58G8HT4 |
| grw1 | **-7.0** | 8.6 | prop | eman | -62.22 | -58.96 | Dietrich et al. (2004) |
| haa1 | **3.9** | 100.0 | prop | eman | -77.04 | -78.29 | British Antarctic Survey |
| haag | **6.1** | 1.1 | rman | eman | -77.04 | -78.29 | doi:10.7283/T5FT8JB8 |
| howe | **0.6** | 1.1 | rman | eman | -87.42 | -149.43 | doi:10.7283/T5ZW1J65 |
| hown | **3.9** | 0.8 | prop | prop | -77.53 | -86.77 | doi:10.7283/T56971WH |
| hton | **4.8** | 3.7 | prop | eman | -74.08 | -61.73 | doi:10.7283/T5222RV6 |
| hugo | **0.9** | 1.3 | prop | prop | -64.96 | -65.67 | doi:10.7283/T5FQ9TW3 |
| iggy | **2.3** | 1.1 | prop | eman | -83.31 | 156.25 | doi:10.7283/T5QC01T9 |
| jnsn | **4.0** | 1.7 | prop | prop | -73.08 | -66.10 | doi:10.7283/T5SJ1HP1 |
| lntk | **4.6** | 3.1 | rman | eman | -74.84 | -73.90 | doi:10.7283/T5J1017P |
| lply | **2.0** | 8.1 | rman | eman | -73.11 | -90.30 | doi:10.7283/T5DV1H50 |
| lwn0 | **2.1** | 1.0 | prop | prop | -81.35 | 152.73 | doi:10.7283/T5T43RD8 |
| mait | **0.4** | 1.1 | rman | eman | -70.77 | 11.74 | Dietrich et al. (2004) |
| mar1 | **7.1** | 10.0 | prop | eman | -64.24 | -56.66 | Dietrich et al. (2004) |
| mbl1 | **2.5** | 3.0 | prop | eman | -78.03 | -155.02 | Donnellan & Luyendyk (2004) + doi:10.7283/T5CJ8BS7 |



| | | | | | | | |
|---|---|---|---|---|---|---|---|
| mbl2 | **2.3** | 10.0 | prop | eman | -76.32 | -144.31 | Donnellan & Luyendyk (2004) |
| mbl3 | **1.3** | 17.9 | rman | eman | -77.34 | -141.87 | Donnellan & Luyendyk (2004) |
| mcar | **3.7** | 1.4 | prop | prop | -76.32 | -144.30 | doi:10.7283/T55D8Q41 |
| mirn | **24.4** | 100.0 | prop | eman | -66.55 | 93.01 | SCAR |
| mkib | **4.7** | 2.6 | rman | eman | -75.28 | -65.60 | doi:10.7283/T5D798HD |
| mtcx | **-3.8** | 10.0 | prop | eman | -78.52 | 162.53 | Raymond et al. (2004) |
| ohg1 | **4.5** | 10.0 | prop | eman | -63.32 | -57.90 | Dietrich et al. (2004) |
| ohig | **4.0** | 0.7 | prop | prop | -63.32 | -57.90 | Former IGS: Dow et al. (2009) |
| pal1 | **8.1** | 10.0 | prop | eman | -64.77 | -64.05 | Dietrich et al. (2004) |
| patn | **4.8** | 0.7 | prop | prop | -78.03 | -155.02 | doi:10.7283/T5PC30PX |
| pece | **0.7** | 4.2 | prop | eman | -85.61 | -68.56 | doi:10.7283/T5930RG1 |
| pra1 | **4.2** | 10.0 | prop | eman | -62.48 | -59.65 | Dietrich et al. (2004) |
| prat | **-9.6** | 100.0 | prop | eman | -62.48 | -59.65 | doi:10.7283/T5M32T21, doi:10.7283/T5K35RZP |
| prtt | **-5.0** | 100.0 | prop | eman | -62.48 | -59.67 | SCARP*** |
| reyj | **151.3** | 300.0 | prop | eman | -62.20 | -58.98 | doi:10.7283/T5M32T21, doi:10.7283/T5K35RZP |
| rob1 | **5.4** | 5.1 | prop | eman | -77.03 | 163.19 | doi:10.7283/T5057D6V, doi:10.7283/T53T9FHJ |
| robi | **8.7** | 1.5 | prop | prop | -65.25 | -59.44 | Nield et al. (2014) |
| rot1 | **6.5** | 10.0 | prop | eman | -67.57 | -68.13 | SCAR |
| rotb | **5.0** | 0.4 | prop | prop | -67.57 | -68.13 | doi:10.7283/T56M34Z7 |
| roth | **5.5** | 1.4 | prop | prop | -67.57 | -68.13 | IGS: Dow et al. (2009) |
| sdly | **-0.3** | 1.4 | prop | prop | -77.14 | -125.97 | doi:10.7283/T5S46Q7F |
| sig1 | **23.0** | 100.0 | prop | eman | -60.71 | -45.59 | Dietrich et al. (2004) |
| smr1 | **0.5** | 10.0 | prop | eman | -68.13 | -67.10 | Dietrich et al. (2004) |
| smrt | **1.2** | 0.9 | prop | prop | -68.13 | -67.10 | Alfred Wegener Institute / Instituto Antartico Argentina |
| sppt | **12.9** | 100.0 | prop | eman | -64.29 | -61.05 | Bevis et al. (2009) |
| sugg | **4.7** | 1.3 | rman | eman | -75.28 | -72.18 | doi:10.7283/T5CV4G1M |
| svea | **1.3** | 1.1 | prop | prop | -74.58 | -11.23 | Sjoberg et al. (2011) |
| thur | **-1.2** | 2.5 | rman | eman | -72.53 | -97.56 | doi:10.7283/T5862DRZ |
| tomo | **47.7** | 20.3 | rman | eman | -75.80 | -114.66 | doi:10.7283/T5BZ64B0 |
| trve | **2.5** | 5.6 | rman | eman | -69.99 | -67.55 | doi:10.7283/T5NS0RZ9 |
| ver1 | **0.3** | 100.0 | prop | eman | -65.25 | -64.26 | SCAR |
| ver3 | **-6.2** | 100.0 | prop | eman | -65.25 | -64.26 | SCAR |
| vnad | **4.4** | 1.1 | prop | prop | -65.25 | -64.25 | doi:10.7283/T52F7KQ1 |
| w01b | **1.4** | 10.0 | prop | eman | -87.42 | -149.44 | |
| w02b | **2.3** | 10.0 | prop | eman | -85.61 | -68.56 | doi:10.7283/T5445JTQ |
| w03a | **-1.4** | 10.0 | prop | eman | -81.58 | -28.40 | doi:10.7283/T50C4T3D |
| w03b | **1.7** | 10.0 | prop | eman | -81.58 | -28.40 | |
| w05a | **2.3** | 10.0 | prop | eman | -80.04 | -80.56 | doi:10.7283/T57W69HP |
| w05b | **7.4** | 10.0 | prop | eman | -80.04 | -80.56 | doi:10.7283/T50C4T3D |



| w06a | **-2.2** | 100.0 | prop | eman | -79.63 | -91.28 | |
|------|------|-------|------|------|--------|--------|---|
| w07a | **3.3** | 100.0 | prop | eman | -80.32 | -81.43 | |
| w08a | **-1.5** | 100.0 | prop | eman | -75.28 | -72.18 | |
| w09a | **2.2** | 100.0 | prop | eman | -82.68 | -104.40 | |
| wasa | **0.6** | 3.2 | prop | eman | -73.04 | -13.41 | Sweden |
| whn0 | **2.2** | 0.9 | prop | prop | -79.85 | 154.22 | doi:10.7283/T5R49P2M |
| whtm | **7.7** | 0.8 | prop | prop | -82.68 | -104.39 | doi:10.7283/T5ZP44DZ |
| wiln | **4.9** | 0.9 | prop | prop | -80.04 | -80.56 | doi:10.7283/T53F4MX9 |

*https://www.unavco.org/projects/project-support/polar/geodetic/benchmarks/sites/crar.html (accessed 1 June 2017)

**Geoscience Australia GNSS archive at ftp://ftp.ga.gov.au/geodesy-outgoing/gnss/ as of 1 June 2017. See also Brown, N. and Woods, A., 2008. Antarctic Geodesy 2006 – 2007 Field Report. Geoscience Australia, Record 2009/32. 77pp.

*** SCARP Campaign datasets, doi:10.7283/T5T151QB, doi:10.7283/T59P2ZZD, doi:10.7283/T5K35RZP. Also see https://gcmd.nasa.gov/records/GCMD_JCADM_USA_SCARP.html




Table A.3. Comparison of 'prop,eman' GPS uplift rates for this study with rates from other studies.

| | REGINA | | Thomas et al. 2011 | | Argus et al. (2014) | | Wolstencroft et al. (2015) | |
|---|---|---|---|---|---|---|---|---|
| | $y^u$ | $\sigma^u$ | $y^u$ | $\sigma^u$ | $y^u$ | $\sigma^u$ | $y^u$ | $\sigma^u$ |
| a351 | -0.9 | 1.8 | 0.8 | 1.3 | 1.1 | 3.5 | | |
| a368 | -0.2 | 1.2 | 0.4 | 1.0 | | | | |
| for1 | -0.2 | 2.9 | -1.4 | 0.8 | | | | |
| for2 | -0.3 | 2.7 | 2.1 | 0.9 | | | | |
| fos1 | 3.1 | 1.3 | 2.1 | 0.4 | 2.9 | 1.2 | 3.9 | 1.1 |
| ftp1 | -2.2 | 3.4 | 2.1 | 2.8 | | | | |
| hton | 4.8 | 3.7 | | | | | | |
| mbl1 | 2.5 | 3.0 | 0.6 | 1.5 | | | | |
| mbl2 | 2.3 | 10 | 0.2 | 4.1 | | | 6.4 | 0.9 |
| rob1 | 5.4 | 5.1 | 7.5 | 2.6 | | | | |
| w01a(-howe) | -0.3 | 10 | -2.5 | 1.7 | 0.9 | 1.2 | | |
| w01b | 1.4 | 10 | -3.1 | 1.7 | | | | |
| w02a(-pece) | 0.3 | 10 | 2.8 | 1.2 | −1.2 | 1.9 | | |
| w02b | 2.3 | 10 | 0.5 | 1.9 | | | | |
| w03a | -1.4 | 10 | -3.2 | 1.8 | −1.1 | 2.4 | | |
| w03b | 1.7 | 10 | -1.7 | 1.8 | | | | |
| w04a | 3.7 | 100 | 3.0 | 1.1 | | | | |
| w05a | 2.3 | 10 | 3.5 | 2.0 | | | | |
| w05b | 7.4 | 10 | 5.3 | 1.2 | | | | |
| w06a | -2.2 | 100 | -2.2 | 2.4 | −4.7 | 4.4 | | |
| w07a | 3.3 | 100 | 3.3 | 2.1 | 4.6 | 3.1 | | |
| w08a(b/sugg) | -1.5 | 100 | 1.3 | 1.3 | | | | |
| w09a | 2.2 | 100 | 4.5 | 2.6 | | | | |



*A.5 Choice of GRACE cut-off degree and biasing*

In this study, we identify GRACE coefficients of CSR RL05 up to-degree and order 50 appropriate to yield the most robust gravity field rates over Antarctica. Figure A.3 provides

another indication based on the degree-power spectrum of the geoid rates. It is visible that GFZ RL05 and CSR RL05 are very similar up to degree and order 50, where the power spectra show minima. For higher degrees, however, the power of the gravity field recovered with GRACE

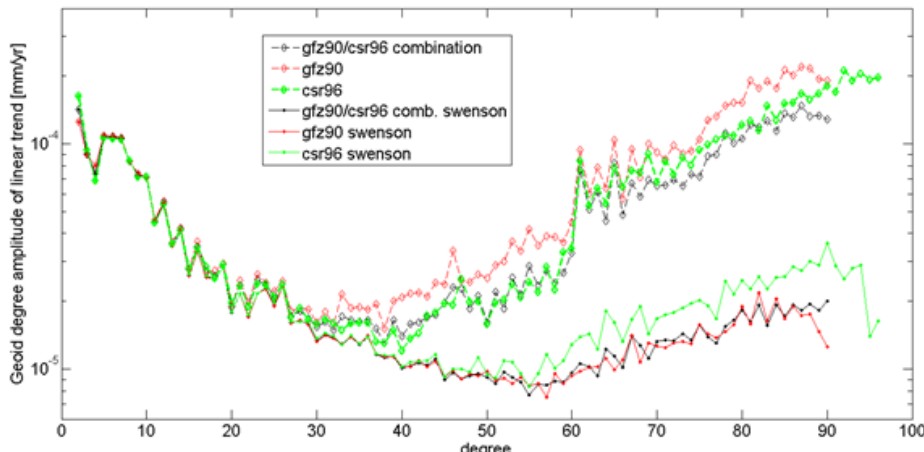

*Figure A.3: Degree-amplitude spectrum of the rate of geoid-height change (mm/yr) for unfiltered*

*(diamond-dashed lines) and for Swenson-filtered (solid lines) solutions. Red: GFZ; green:*

*CSR; black combination of GFZ and CSR with equal weights.*

increases due to increasing noise, for the unfiltered coefficients particularly faster for GFZ RL05 than for CSR RL05.

The filtering of the GRACE gravity fields was optimized for reducing noise over

Antarctica. The effect on the RMS uncertainties is shown in Fig. 3. Additionally, Fig. A.4



presents the difference of between the GRACE rates filtered only with a Gaussian smoothing
filter of 200 km, and additionally with the optimized Swenson filter. It is visible that the
differences in the rate of geoid-height change and the associated rate of equivalent water-height
change, respectively, show a stripe-like noise pattern. This suggests that the de-striping is
superior over conventional Gaussian smoothing, even at high latitudes, where GRACE ground-
track spacing is very dense. It is also important to note that the filter does not introduce any
magnitude bias, or changes the spectral content of the gravity field rates, which is important
when applying only Gaussian smoothing of 200 km (without Swenson filtering) to the altimetry
data set and response kernels.

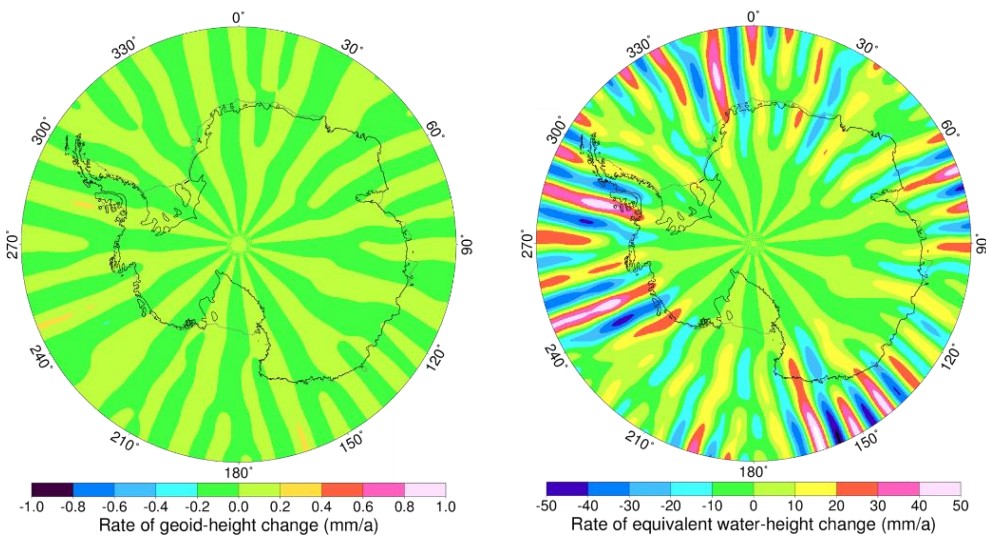

*Figure A.4: Spatial rate of geoid-height change (left) and rate of equivalent water-height change*
*(right) (mm/yr) for the difference between the GRACE trends processed by Gaussian*
*smoothing of 200 km and the optimal Swenson filter & Gaussian smoothing. The solutions*
*are CSR RL05, the spherical-harmonic cut-off degree is 50.*



### A.6 Evaluation of assumption of viscoelastic equilibrium state

The viscoelastic response kernels employed (Section 5) describe the viscoelastic

equilibrium state for the forcing with a disc load of constant radius and constant rate of mass

increase (likewise mass loss). We neglect transitional changes of the solid Earth for load

changes that have not reached the equilibrium state in terms of geoid-height change and surface

displacement. Although, the deformation and gravity signature in equilibrium eventually only

depends on the lithosphere thickness, the time to reach the equilibrium is controlled by the

viscosity parameters chosen. Fig. A.5 shows the evolution of the standardized ratio of the geoid-

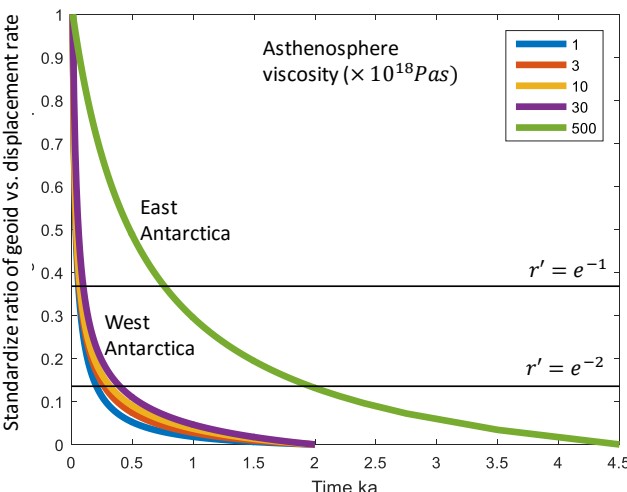

*Figure A.5: Standardized ratio of the rate of geoid-height change versus the rate of radial displacement for different values of the asthenosphere viscosity. Note that the ratio is calculated at the load center.*

height change vs. surface displacement over time, calculated as

$r' = [r(t) - r(t = t_{tmax})] / \max[r(t) - r(t = t_{tmax})])$,    where    $r = y^g(t)/y^u(t)$    is



evaluated at the load centre. It is visible that for the weaker West Antarctic rheology

(asthenosphere viscosity between $1 \times 10^{18}$ Pa s and $3 \times 10^{19}$ Pa s) $r'$ falls to $1/e^2$ within the

500 yr. For East Antarctica ($1 \times 10^{20}$ Pas), $r' = e^{-2}$ is reached within 2 kyrs. With this quasi-

942 stationary solution approach, the inference on the timing of the past ice mass change is limited

to an upper limit in terms of magnitude, and a lower limit in terms of load duration; a similar

ratio is achieved by a thinner lithosphere thickness, which has not reached viscoelastic

equilibrium state, and earlier load changes are fully relaxed, respectively.

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
