# Peer review of "Altimetry, gravimetry, GPS and viscoelastic modelling data for the joint inversion for glacial isostatic adjustment in Antarctica (ESA STSE Project REGINA)"

_Earth System Science Data, 2017_

## Referee Comment (RC1) · Anonymous Referee #1 · 18 Aug 2017

Geodetic measurements in Antarctica measure a combination of Glacial Isostatic Adjustment (GIA) and snow and ice thickness changes in Antarctica. Combination of the different data sets in an inversion approach might be the best method to isolate the different components. Such inversion imposes requirements on the data sets. This paper presents analysis of data sets (altimetry, GPS, satellite gravity) and GIA model outputs to convert between the observables. The products can be used in an inversion to separate the different components which is done in a separate study. However, the data sets can also be a useful resource for studies relying on one of the data sets. It is commendable that the authors have put great care in processing the data and making the results available. It will be a very useful resource for Antarctic mass balance studies. I reviewed an earlier version of the manuscript and I appreciate that comments from that review have been addressed in the current manuscript. There are in my opinion still several minor issues related to the description. The paper does not make sufficiently clear in the introduction what the processing adds to previous studies and what is required of the data sets to be used in the inversion in paper II. Such explanation would guide the reader of this lengthy paper. Given that the main aim is to present 'data inputs', the descriptions of processing and errors is sometimes ambiguous. I hope the specific comments below help to improve this.

Specific comments

74: the statement that forward models overpredict uplift rate measurements is not generally true, there are regional models that are tuned to the GPS data and there are instances where standard models underpredict observed uplift rates, see Wolstencroft et al (GJI 2015)

102: How are the response functions used in combining them?

Introduction: The introduction states that the data sets and modelling results are of value to address other research questions. But the paper itself does not yet contain a research question. In addition, it is not clear form the introduction why the processing is better than previous analysis of the data sets. For example, would you expect improvement compared to Thomas et al. (2011) or are differences merely 'small processing strategy changes' (line 272). It should also be summarized what requirements the inversion poses on the data and kernels, for example in terms of time period, resolution, error (= weight in the inversion). Such explanation would help the reader evaluate the (many) choices that are made in the manuscript.

121 and further. More information is given on the corrections, which is helpful, but not yet what the error in the corrections is (or if it is insignificant) and if it is added to the

height error.

138: 'residual uncertainties' is confusing as it sounds like the residual of the uncertainties? In any case it does not correspond to equation 1, which gives non-dimensional values as both e and x have the same unit. Also, it should be discussed why residuals are a good approximation for errors.

160: 'the standard deviations of the rates'. Are they also calculated according to equation 1?

208: Errors could be important in the inversion to weigh the contribution from the different data sources. Neglecting model uncertainties because estimates are not available is not really a satisfactory solution.

243: This is the first time this data set is mentioned. Does it include error estimates?

339: Do you have any explanation for the difference?

361: What is the threshold and how did you weigh the average? This manuscript present data sets and their analysis so the procedure should be clear.

470: it is not clear what is meant. Is the search range for the parameters limited? Is the range of m limited to values higher than 10?

476 and further, it is confusing to use both interannual and non-linear because they can seem the same but are not necessarily so.

404 "zero difference": better to write a full sentence here.

495: "the post-fit RMS residual for this known temporal signal variation". This is not clear. In line 449 the residual is defined as GRACE minus ice elevation, fitting is not mentioned. figure 5: the axis label states 'linear trend residuals', but the text in page 503 states that also annual oscillations are removed. Please make the descriptions consistent.

509 and further. The procedure seems OK but the reasoning does not make sense. if you downweigh months with high post-fit RMS the post-fit RMS decreases. That seems to me a mathematical certainty and in that case it should not be used to say that the downweighting is beneficial.

514: What is meant by more accurate? A higher RMS when you include noisier months is still an accurate representation of the noise.

Section 4: it is not clear to me what is done with the signal corruption due to Swenson and Gaussian filtering. Is that added to the error? Or will the filtering be applied to the other datasets in the inversion? Line 924 states that there is no magnitude bias (in the geoid rate?), but that would suggest that filtering is not really necessary

594: it would help the reader to be more clear about why you need the response kernels in the inversion. Only in the conclusions on line 846 it is mentioned that you need the kernels for ratio of gravity and displacement.

631: Does the range span the values in the Priestley and McKenzie 3D viscosity model that you use later?

645: $10^{22}$ is quite low to be considered fully elastic. Such viscosity would still give noticeable response from ice load changes since the last glacial maximum.

657: make clear that it is the standardized ratio (i.e. it starts at 1)

658: according to appendix A.6 it should be $1/e^2$

section 5.5: another assumption(mentioned in the appendix in line 932) is that the equilibrium has been reached. If load changes constantly, then at present you are not in a state of equilibrium with constant displacement rate. This is mentioned later but could also be added here. Another assumption is that upper and lower mantle viscosity are assumed known.

733: e-dot was used for geoid rate in line 673

842: The response functions in the paper are produced for a continuously changing load. It is not yet possible to draw conclusions about the exact timing of the load from that.

846: the ratio should be for rates, not the gravity disturbance itself.

848 and further: this is an important justification that should be mentioned in the introduction as well

935: and on elastic parameters and density

Typos etc

81: grammar 'And thus to'

95: grammar, change 'invasion'

115: I suggest adding this to the acknowledgements instead

150: 'the' before ICESAT

figure 1: when zooming in I see many different colors. That might be the result of lower resolution picture, but it makes it hard to interpret the colors described in the caption

caption figure 2: space before sigma

213: should it be 20 km grid?

219: typo? something wrong with the degree symbol here and further on

228: abbreviation should be introduced

229: typo?

246: kg/mˆ3 instead of km/mˆ3

302: can refer to section 3.2.2

caption Table 4: "Table Appendix A.4"

370: provide link?

533: expanded 'to'

593: remove 'a'

601: add 'are' before 'a classic'

648: considered

649: parameter

801: compositing?

807: terms

813: change 'over' to 'about'

834: 'however' implies a contradiction, I don't see that

table A.2, better to write approx in full.

figure A.4 and text use both mm/a and mm/year

figure A.5 axis label: standardized

---

## Referee Comment (RC2) · Anonymous Referee #2 · 16 Oct 2017

This paper gives a detailed description of a group of linked geodetic datasets for Antarctica: ice surface height change from altimetry, gravity change from GRACE, and uplift rates from GPS. In addition, a set of viscoelastic response functions are provided, which can be used for joint inversion or other modeling of these data. The paper is comprehensive and thorough, and with a few minor points of confusion, gives a complete description of these datasets. The datasets themselves are useful, and although some version of each has been published before, there is good reason for people to use these latest and more complete versions. The authors do an excellent job in description and complete citation, and are to be commended for including the DOIs for the GPS sites that they use.

The paper seems to indicate that the code (in some cases) is also provided (and function names and so on are provided). But I do not see the code in the repository. I encourage the authors to include the code with the data. There was also mention in the paper of some other products such as the geodesic grid, which at present are not included in the repository. I see the value in including this, so the authors need to carefully check the repository for completeness. There also are some intermediate (calibration) data products that might be worth considering to add to the repository, in particular the time series for the firn correction and the SMB (as de-trended for this study).

I marked a number of minor corrections on an annotated manuscript. Only the pages with corrections are included. More substantial comments are given below.

Introduction: Explain why the 2003-2009 time span was chosen for GRACE. It matches the altimetry, but a longer time span was used for GNSS. Also, emphasize more strongly the key assumption of a constant rate of change of all variables over the time period chosen, especially where datasets do not exactly coincide in time.

Line 158. Units mismatch: "rate" and "cm". cm/year?

Section 2.3. Why not use a weighted average of the two datasets? I can guess that maybe when one of the datasets has a large uncertainty then it can be really bad? Could you either present the values or statistics of ( data_type_chosen – data_type_not_chosen)/(sqrt(sum of sigmas^2)? The reader wonders how well the two altimetry sources agree.

Section 2.4 Explain how the 27 km grid of firn corrections is re-sampled to the 10x10 km grid used for the altimetry. Standard bilinear interpolation, or something else? Also, line 213 refers to a 20 km grid. What is the relationship between this grid and the 10x10

grid? Are the 10x10 grid values decimated to 20 km? Block mean, block median? Or is the reference to 20 km an error in the text?

Section 3. Why not just use JPL's orbits and clocks? A short explanation is enough. I presume that the reason relates to ensuring consistency.

Line 281. Give the median values for white noise sigma scaling and the flicker noise amplitude.

In all the tables in Section 3, please clarify if the Argus et al. (2014) result are as originally presented, or if you attempted to correct for the small frame difference.

Lines 617-618. This sentence says that the loading rate is held constant until dynamic equilibrium is reached, which as written suggests that the loading rate is changed after that time. How is it changed? Or is this simply an error in English? Based on other parts of the text, it seems that the loading rate was actually held constant for 2000 years (West Antarctica) or 15,000 years (East Antarctica), and that after this an extra time step is done with no loading change to give the purely viscoelastic response. In fact, the 2000 years and 15,000 years are argued to be longer times than needed to reach dynamic equilibrium, if I understood the authors properly.

Line 658. Clarify whether you mean "to a value of 1/e" (as written), or "by a factor of 1/e relative to the initial value".

Fore figure 7-11, it would be useful to complement the versions shown with a figure that shows the uplift rate as a function of time at the center of the load. If there is any change in the loading rate during the calculation (see the comment on the ambiguity introduced by lines 617-618), then the loading rate should also be shown.

SMB correction. Because the SMB correction for the GRACE has been detrended over the GRACE time period, it does not contribute to the rate. However, the SMB variations might contribute to the GPS uplift rates given that the time spans of these data vary. I wonder if this effect has been considered and whether it might be

important? Sites with strongly non-linear time series were identified and excluded or handled separately, but there may be some subtle biases that remain. I think it would be useful to evaluate this impact, or at least to provide the de-trended SMB grid time series so that it could be assessed.

Please also note the supplement to this comment: https://www.earth-syst-sci-data-discuss.net/essd-2017-46/essd-2017-46-RC2-supplement.pdf

---

## Author Comment (AC1) · 12 Nov 2017

Geodetic measurements in Antarctica measure a combination of Glacial Isostatic Ad-justment (GIA) and snow and ice thickness changes in Antarctica. Combination of the different data sets in an inversion approach might be the best method to isolate the dif-ferent components. Such inversion imposes requirements on the data sets. This paper presents analysis of data sets (altimetry, GPS, satellite gravity) and GIA model outputs to convert between the observables. The products can be used in an inversion to sep-arate the different components which is done in a separate study. However, the data sets can also be a useful resource for studies relying on one of the data sets. It is commendable that the authors have put great care in processing the data and making the results available. It will be a very useful resource for Antarctic mass balance studies. I reviewed an earlier version of the manuscript and I appreciate that comments from that review have been addressed in the current manuscript. There are in my opinion still several minor issues related to the description. The paper does not make sufficiently clear in the introduction what the processing adds to previous studies and what is re-quired of the data sets to be used in the inversion in paper II. Such explanation would guide the reader of this lengthy paper. Given that the main aim is to present 'data inputs', the descriptions of processing and errors is sometimes ambiguous. I hope the specific comments below help to improve this.

General comments:

The paper does not make sufficiently clear in the introduction what the processing adds to previous studies and what is re-quired of the data sets to be used in the inversion in paper II

We have added a description of the requirements imposed by the joint inversion. Moreover, we now describe the benefit of using response functions for the inversion at several places in the text. The processing of the data sets is not revolutionary, but the consistency and refinements achieved was only possible by the effort and discussions of the individual data providers engaged in the project.

Given that the main aim is to present 'data inputs', the descriptions of processing and errors is sometimes ambiguous

We have resolved issue of ambiguous processing and error descriptions, also raised by Reviewer 2.

Specific comments

74: the statement that forward models overpredict uplift rate measurements is not generally true, there are regional models that are tuned to the GPS data and there are instances where standard models underpredict observed uplift rates, see Wolstencroft et al (GJI 2015)

That is true. Another example of under-predicted GIA is the large uplift rates in the Amundsen Sea Embayment. We modified the statement. [R1_076]

102: How are the response functions used in combining them?

We added a clarifying text why the response functions are of advantage for solving the joint inversion problem based on the different input data (i.e. geophysical meaningful and Earth structure dependent ratio of the rate of geoid change / surface displacement, possible to use different filters on the input data sets). [R1_105]

Introduction: The introduction states that the data sets and modelling results are of value to address other research questions. But the paper itself does not yet contain a research question. In addition, it is not clear form the introduction why the processing is better than previous analysis of the data sets. For example, would you expect improve-ment compared to Thomas et al. (2011) or are differences merely 'small processing strategy changes' (line 272).

We specified the most important advancement in the processing of each data set [R1_099].

It should also be summarized what requirements the in-version poses on the data and kernels, for example in terms of time period, resolution, error (= weight in the inversion). Such explanation would help the reader evaluate the (many) choices that are made in the manuscript.

We spelled out the requirements necessary for the joint inversion [R1_111]

121 and further. More information is given on the corrections, which is helpful, but not yet what the error in the corrections is (or if it is insignificant) and if it is added to the height error.

We did not evaluate the uncertainties caused by variations in the processing choices. This is now stated in the text [R1_157].

> 138: 'residual uncertainties' is confusing as it sounds like the residual of the uncertainties? In any case it does not correspond to equation 1, which gives non-dimensional values as both e and x have the same unit. Also, it should be discussed why residuals are a good approximation for errors.

We reformulated the term "residual uncertainties". We referred the reader to Eq. 1 in Hurkmans et al. 2012, who provides a detailed description of the error estimation. And we explained why residuals are used as part of the uncertainty estimate [R1_168].

> 160: 'the standard deviations of the rates'. Are they also calculated according to equation 1?

Yes. We included a reference to Eq. 1.

> 208: Errors could be important in the inversion to weigh the contribution from the different data sources. Neglecting model uncertainties because estimates are not available is not really a satisfactory solution.

We agree that the uncertainties should in principle be considered. But they will be small, such that the current estimate of elevation rate uncertainties captures the dominant sources. We added explanatory sentences [R1_243]

> 243: This is the first time this data set is mentioned. Does it include error estimates?

Yes, we added this data set for completeness, although it is discussed in Sasgen et al. 2017. We do not provide uncertainties with it [R1_284].

> 339: Do you have any explanation for the difference?

Currently, this difference is unexplained. A step-by-step intercomparison of the GPS processing with Wolstencroft et al. 2015 beyond the scope of this paper. We added the sentence "The systematic differences between Wolstencroft et al. (2015) and the REGINA values for Palmer Land are currently unexplained and a matter of ongoing investigations" [R1_370]

> 361: What is the threshold and how did you weigh the average? This manuscript present data sets and their analysis so the procedure should be clear.

We added the thresholds for the clustering, and specified that weighted averaging was done for the positions. [R1_361]

> 470: it is not clear what is meant. Is the search range for the parameters limited? Is the range of m limited to values higher than 10?

We added clarifying text. [R1_470].

471: 476 and further, it is confusing to use both interannual and non-linear because they can seem the same but are not necessarily so.

We now consistently use "non-linear" now instead of "interannual". The positions in the text are marked [R1_476].

404 "zero difference": better to write a full sentence here.

We added a full sentence. [R1_404]

495: "the post-fit RMS residual for this known temporal signal variation". This is not clear. In line 449 the residual is defined as GRACE minus ice elevation, fitting is not mentioned. figure 5: the axis label states 'linear trend residuals', but the text in page 503 states that also annual oscillations are removed. Please make the descriptions consistent.

We changed the labeling and caption of Fig. 5. And, added clarifying text. The optimization of the filter parameters is done on GRACE rates minus ice elevation rates. The SMB reduction is intended to reduce multi-year fluctuations, for improving the residual GRACE uncertainty – i.e. to see whether the residual uncertainty gets closer to the nominal calibrated uncertainty of the GRACE coefficients. The de.trending of step 3 includes a annual oscillation to account for remaining seasonal variations of SMB not captured by the SMB model. The seasonal is not important and could be neglected for this analysis yielding the similar results [R1_495].

509 and further. The procedure seems OK but the reasoning does not make sense. if you downweigh months with high post-fit RMS the post-fit RMS decreases. That seems to me a mathematical certainty and in that case it should not be used to say that the downweighting is beneficial.

Accepted. The second half of the sentence was removed. [R1_509]

514: What is meant by more accurate? A higher RMS when you include noisier months is still an accurate representation of the noise.

We changed the wording [R1_514].

Section 4: it is not clear to me what is done with the signal corruption due to Swenson and Gaussian filtering. Is that added to the error? Or will the filtering be applied to the other datasets in the inversion? Line 924 states that there is no magnitude bias (in the geoid rate?), but that would suggest that filtering is not really necessary

The signal corruption itself is not considered as a component of the uncertainty estimate. It is arguable, whether the Swenson filtering should be applied to the altimetry data before the joint inversion, as this data set has a different error structure. The largest effect is due to the signal loss caused by the smoothing with a Gaussian filter. This is considered by applying the Gaussian smoothing to the altimetry data set and the viscoelastic response functions. We added more explanatory text [R1_511].

594: it would help the reader to be more clear about why you need the response kernels in the inversion. Only in the conclusions on line 846 it is mentioned that you need the kernels for ratio of gravity and displacement.

We added explanations in the introduction [see R1_105]. We also added more explanation in Section 5 [R1_653].

631: Does the range span the values in the Priestley and McKenzie 3D viscosity model that you use later?

Yes, the range is guided by the model of Priestley & McKenzie (2013). We added a sentence. [R1_696]

645: 10ˆ22 is quite low to be considered fully elastic. Such viscosity would still give noticeable response from ice load changes since the last glacial maximum.

We agree. However, the value has to be considered as a threshold value to map the continuous viscoelastic parameter in Priestley & McKenzie (2013) obtained by Earth modelling to our layered model for the calculating the viscoelastic relaxation. The effect of changing this parameters was presented in Sasgen et al. 2017 (Fig. S4). Explanation was added. [R1_712].

657: make clear that it is the standardized ratio (i.e. it starts at 1)

Clarified [R1_700].

658: according to appendix A.6 it should be 1/eˆ2

Corrected [R1_700].

section 5.5: another assumption(mentioned in the appendix in line 932) is that the equilibrium has been reached. If load changes constantly, then at present you are not in a state of equilibrium with constant displacement rate. This is mentioned later but could also be added here. Another assumption is that upper and lower mantle viscosity are assumed known.

We agree. We added these assumptions to the list [R1_795].

733: e-dot was used for geoid rate in line 673

We unified the nomenclature.

842: The response functions in the paper are produced for a continuously changing load. It is not yet possible to draw conclusions about the exact timing of the load from that.
843:

Added a sentence on this limitation [R1_921]

846: the ratio should be for rates, not the gravity disturbance itself.

Agreed. Added "rates of". [R1_923].

848 and further: this is an important justification that should be mentioned in the introduction as well

We added the justification to the Introduction and to the Section 5.

935: and on elastic parameters and density

Typos etc

81: grammar 'And thus to'

Corrected.

95: grammar, change 'invasion'

Corrected.

115: I suggest adding this to the acknowledgements instead

Moved to the acknowledgements.

150: 'the' before ICESAT

Inserted 'the'.

figure 1: when zooming in I see many different colors. That might be the result of lower resolution picture, but it makes it hard to interpret the colors described in the caption

This seems to be a resolution problem. We will make sure this is ok in the final digital version.

caption figure 2: space before sigma

Inserted space.

213: should it be 20 km grid?

The grid resolutions as stated are correct.

219: typo? something wrong with the degree symbol here and further on

Corrected degree symbols.

228: abbreviation should be introduced

FDM is now spelled out "firn-densification model".

229: typo?

Corrected.

246: kg/m^3 instead of km/m^3

Very true. Thank you.

302: can refer to section 3.2.2 caption

Table 4: "Table Appendix A.4"

Reference included

370: provide link?

CS

**Resolved missing link.**

533: expanded 'to'

**Inserted 'to'**

593: remove 'a'

**Removed.**

601: add 'are' before 'a classic'

**Included 'is'**

648: considered

**Corrected.**

649: parameter

**Changed to singular.**

801: compositing?

**Removed.**

807: terms

**Corrected.**

813: change 'over' to 'about'

**Changed.**

834: 'however' implies a contradiction, I don't see

**Removed.**

table A.2, better to write approx in full.

**Changed.**

figure A.4 and text use both mm/a and m/year

**Changed.**

figure A.5 axis label: standardized

**Labels unified.**

[revised manuscript text omitted]

recover optimal temporal linear trends. [R2_020] We focus on the trends derived for the time

period 2003-2009 in which GRACE and ICESat operated simultaneously. Note that the

96   stationarity of the trend is a key assumption underlying our approach, when including GPS rates

covering a longer time span (1995-2013.7). However, limiting the GPS data to the time span

2003-2009 leads to a significant reduction of the number of stations for which reliable trends

99   can be estimated, and, hence, a loss of spatial coverage. For comparison, the reader is advised

to the data archive, in which GPS uplift rates for the time periods 2003-2009 and 2003-2013.7

are made available.

102   [R1_099] In this paper, we present refined We refine existing procedures for estimating

trends for of the data sets on surface-ice elevation changes, surface displacement and gravity

field changes. The rates of surface-ice elevation changes from Envisat and ICESat satellite

105   altimetry are improved by (Section 2), by combining both data sets based on their respective

uncertainties, increasing the spatial coverage and accuracy of the elevation rates (Section 2). ;

bedrock Bedrock displacement –from *in situ* networks of GPS stations in Antarctica are

108   improved in coverage by allowing for campaign-based data and carefully assessing the

uncertainty of the trend with a noise model (Section 3). Compared to the rates in Thomas et al.

(2011) also more stations and longer time series are included, and The gravity field changes

[7]

111    from GRACE are refined compared to previous work by optimizing the de-striping filtering for the region of Antarctica (Section 4). [R1_111] The processing aims at fulfilling the requirement of the joint inversion to combine input data based on the same time period (not possible for

114    GPS without having to ignore a large number of stations) and covering entire Antarctica, accompanied by a realistic description of the uncertainties.

We also present forward modelling results of viscoelastic response functions to disc load

117    forcing for the range of Earth structures likely to prevail in Antarctica (Section 5). [R1_105] The viscoelastic response functions allow us to combine the surface displacement and gravity changes based on the physical description of the Earth's viscoelastic response for a specified

120    Earth structure. In addition, the response functions enable us to combine data sets of different spatial resolutions, as this is the case for GPS, GRACE and altimetry.

The determination of viscoelastic response functions is a classic topic in solid Earth

123    modelling (e.g. Peltier & Andrews, 1976), though uncommon in the application to joint  inversion studies of satellite data. Although this paper focusses on Antarctica, the response functions and data processing techniques presented here are applicable to other regions. The

126    response kernels represent a wide range of Earth structures and can be used for the separation of superimposed present-day (elastic) and past (viscoelastic) signatures of mass change in other regions with a similar Earth, for example hydrological storage changes and GIA in North

129    America and Alaska. The response functions give insight into the temporal and spatial scales of deformation expected for Antarctica, and are crucial when combining the input data streams.

The data sets and modelling results presented in this paper are accessible in the Pangeae

132    archive, https://doi.pangaea.de/10.1594/PANGAEA.87574 – subsections provide user guidance and point to data and code stored in the archive. As mentioned above, the data sets and

[8]

modelling results are of value to address other research questions as well. For example, the GPS

135 rates provided are useful for the validation of forward modelling GIA solutions, the GRACE

gravity rates can be used for mass balance studies, and altimetry data 2003-2009 can be

extended with the ongoing CryoSat-2 mission to infer volumetric mass balances, also over the

138 ice shelves. The viscoelastic response functions are based on Earth model parameters

potentially suitable to other geographical regions, as well; they are useful for similar studies

combining different data sets of geodetic observables, surface deformation, gravity field

141 change, and topographic change in glaciated areas.

The actual method of the joint inversion is described in a second contribution of the

REGINA project team (Sasgen et al. submitted2017). In this second paper, the resulting GIA

144 estimate is also compared to previous studies. The processing of the data issued here was

enabled by the European Space Agency within the CryoSat+ Support To Science Element Study

REGINA.

147

**2. ALTIMETRY DATA ANALYSIS**

**2.1 *ICESat elevation rate determination**

150    We use along-track altimetry measurements from *ICESat 633 Level 2*, providing high-resolution elevation change observations for the period February 2003 until October 2009. Two corrections are applied to this data set: the range determination from Transmit-Pulse Reference-

153    Point Selection (Centroid vs. Gaussian) (Borsa et al. 2014) available from the National Snow and Ice Data Center (NSIDC), and the inter-campaign correction (Hofton et al. 2013). The Centroid-Gaussian correction is a well-established correction and has been incorporated to the

156    latest ICESat release (634). Concerning the ICESat Intercampaign Bias (ICB) correction, uncertainties are available at Hofton et al (2013). Furthermore, several studies have determined this correction from different methodologies. For a summary of published ICESat ICB

159    corrections see Scambos & Shumman (2016). [R1_157] Note that these corrections are part of a widely accepted procedure and their effect on the elevation rates and uncertainties caused by varying the processing choices have not been evaluated. Because the ICESat tracks do not

162    usually overlap, a regression approach is used in which topographic slope (both across-track and along-track) and the rate of surface-elevation change $y^h_{\text{ICESat}}$, are simultaneously estimated using the 'plane' method (Howat et al. 2008) over areas spanning 700 m long and few hundred

165    meters wide.  A regression is only performed if a plane has at least 10 points from four different tracks that span at least one year. Regression was carried out twice; first, individual elevation measurements with corresponding residuals outside the range of two standard deviations were

168    detected, then, the regression was repeated omitting these outliers. The standard deviation of the regression coefficient, here taken as the uncertainty of the elevation rate, $\sigma^h$(here, ICESat) is calculated by the propagation of the [R1_168] residuals of the uncertainties of the input data

[10]

171  and the estimated topographic heights,

$$\hat{s}_{\text{ICESat}} = \sqrt{\frac{\sum e_i^2 / (n-2)}{\sum (x_i - \overline{x})^2}} \ , \ (1)$$

to the trend parameter (see Eq. 1 in Hurkmans et al. 2012), where $\boldsymbol{e}$ is the vector of

174  residuals, $n$ is the sample size ($i = 1, 2, \ldots, n$), and $\boldsymbol{x}$ is the vector of input elevations with mean

$\overline{\boldsymbol{x}}$. This standard deviation ($\sigma_{\text{ICESat}}$) takes into account the sample size and the variance of both

input data and residuals of the regression (Hurkmans et al. 2012). The residuals of the regression

177  are used as they quantify the approximation of fitting the data with a plane. The exact ICESat

observation periods are shown in the Appendix (A.1, Table A.1). Then, the elevation rate and

its uncertainty are interpolated (bi-linear) [R2_156] to a common $10 \times 10$ km grid in polar-

180  stereographic projection (central latitude 71°S; central longitude 0°W, and origin at the South

Pole, WGS-84 reference ellipsoid).

2.2
*Envisat elevation rate determination*

183  We use a time series of elevation changes derived from along-track Envisat radar altimetry

data for the interval January 2003 to October 2009 (coeval to ICESat time span). Elevation

rates $y_{\text{Envisat}}^{h}$ are obtained at points every 1 km along track, by binning all the echoes within a

186  500 m radius. Then, a 10-parameter least squares model is fitted in order to correct for the

across-track topography and changes in snowpack properties. The least square model is defined

in Flament and Remy (2012). The estimated parameters include parameters determined for the

189  backscatter, leading-edge width and tailing-edge slope, the mean altitude, quadratic surface

slope parameters to define surface curvature and a linear time trend. A digital elevation model

was not used for the correction of the topographic slope. For processing reasons, the temporal

192    resolution is re-sampled from 35 days to monthly periods for each grid cell, before estimating

the elevation rates. This has a minor effect on the elevation rate estimate ([R2_169] smaller

than ± 1 cm/yr) and reduces the standard deviation by about 14 %.  As  with ICESat, the

195    elevation rate is interpolated to common $10 \times 10$ km polar stereographic grid (and $20 \times 20$

km for download in the archive [R2_172]) and the standard deviations of the rates within each

grid cell are taken as an estimate of the measurement uncertainty, $\sigma_{\text{Envisat}}$ according to Eq. 1

198          *Combination of Envisat and ICESat*

2.3    We produce a combined rate of surface-elevation change product from the ICESat and

Envisat datasets for the Antarctic ice sheet, $y^h$. The aim is to take advantage of the high spatial

201    resolution of ICESat data and the high temporal resolution and high-track density of the Envisat

data.

[12]

[Figure]

*Figure 1. Mask for the combination of Envisat/ICESat. ICESat but not Envisat available (yellow),*

$\sigma_{ICESat} \leq \sigma_{Envisat}$ *(green),* $\sigma_{ICESat} > \sigma_{Envisat}$ *(turquoise), Envisat but not ICESat*

*available (orange), and no data (blue). No interpolation is used.*

We combine the two altimetry datasets based on their common $10 \times 10$ km polar-stereographic grid. At each location, the elevation rate with the smallest standard deviation is chosen from either Envisat or ICESat datasets. [R2_181] We prefer this masking procedure instead of a weighted average, in order to avoid introducing possible biases associated with gridded elevation rates of very high uncertainty.

Fig. 1 shows the resulting mask underlying the combination. It is evident that some grid points are only represented by either ICESat or Envisat. Most prominent is the narrowing of the polar gap with ICESat data, resulting from the 81.5°S latitude limit for Envisat compared to 86°S for ICESat due to satellite orbit inclination. On the Antarctic Peninsula, Envisat picks up some points that are not present due to a sparser track coverage in the ICESat data set. As expected, ICESat outperforms Envisat in terms of uncertainty of the elevation rate over steep

topographic slopes and along the ice sheet margins. This is due to the smaller footprint of the laser altimeter, its higher accuracy and lower slope-dependent uncertainty (e.g. Brenner 2007).

216    On some flat areas and over some faulty ground tracks, where ICESat data measurements are scarce, however, Envisat provides better temporal and spatial coverage leading to better accuracy of the resulting elevation rates. The resulting combined data set of surface-elevation

219    rates and its uncertainties are shown in Fig. 2.

[Figure]

*Figure 2: a) Rate of surface-ice elevation change* $y_h$ *and b) associated uncertainties* $\sigma_h$ *derived from Envisat/ICESat combined dataset for the time interval 2003-2009. No interpolation is used; grid points without values are empty (shaded grey).*

*Firn correction*

222  The elevation rates derived from ICESat and Envisat are corrected for changes in the firn

layer thickness using the firn compaction model of Ligtenberg (2011), which is driven by the

regional atmosphere and climate model RACMO2/ANT (Lenaerts, 2010).²·⁴ We determine the

225  firn compaction for January 2003 to October 2009, with respect to the mean of the years 1979

to 2002 and estimate a temporal linear trend, $h_{comp}$. The model output is re-gridded onto the

$10 \times 10$ km common grid using nearest neighbor interpolation. The standard deviation of the

228  re-gridding is less than 1 cm/yr, causing a maximum change of 2 % of the firn compaction rate.

Note that the firn compaction model has a spatial resolution of 27 km, potentially neglecting

finer-scale processes relevant for the altimetry data. Clearly, the re-gridding uncertainty stated

231  above is merely a minimum estimate, neglecting, for example, uncertainties in the calibration

or the atmospheric forcing of the firn compaction model.

The data were re-sampled from every two days to monthly mean time periods for every

234  grid cell before estimating elevation rates. As  with the Envisat and ICESat data, no seasonal

terms  were co-estimated and removed (i.e. annual and semi-annual). We do not apply an *a*

*priori* correction for surface-mass balance (SMB) trends, in accordance with the GRACE

237  processing (Section 5), which requires defining a climatological reference period. Note that

applying the commonly used reference period (1979 to present) leads to spurious accumulation

anomalies in the altimetry data (see Appendix A.2, Fig. A.1). The derivation of an adequate

240  climatological reference epoch in the RACMO2/ANT simulations is in itself challenging and

beyond the scope of this paper.

The total uncertainty of the rate of elevation change from satellite altimetry is calculated

243  by

$$\sigma_h = \sqrt{\sigma_{Envisat/ICESat}^2 + \sigma_{Firn}^2} \quad ,(2)$$

where the standard deviation of the firn correction, $\sigma_{Firn}$ is the formal regression uncertainty (neglecting model uncertainties, as these are not available), and we assume the error sources to be uncorrelated. [R1_243] It is recognized that neglecting uncertainties of the firn model leads to underestimated values of $\sigma_h$. However, the magnitude of the firn correction itself is small (see Appendix A.2) compared to the observational uncertainties, and the associated underestimation of $\sigma_h$ is likely to be small.

**Data availability**

2.5    Annual elevation trends from a combination of Envisat and ICESat data are provided for the time period between February 2003 and October 2009. Trends have been corrected for firn densification processes using RACMO2/ANT.  Elevation trends are provided in a 20 km polar stereographic grid (central meridian 0∘ , standard parallel 71∘ S) with respect to the WGS84 geoid. X and Y are given in km, and the elevation rate and its standard deviation are given in m/yr.

The altimetry data and related ancillary data are directly accessible in the Pangaea

2.5.1    repository: [R2_010]

http://hs.pangaea.de/model/Sasgen-etal_2017/Ice_sheet_topographic_change.zip

*ICESat elevation trend for the* time period between February 2003 and October 2009.

The dataset is provided in a 10 km grid in polar stereographic projection (central meridian 0∘da standard parallel 71∘ S) with respect to the WGS84 geoid. X and Y are given in km, and

[17]

the elevation rate and its standard deviation are given in m/yr.

*Envisat elevation trend for the time period between February 2003 and October*

267   *2009.*

The dataset is provided in a 10 km grid in polar stereographic projection (central meridian

2.5.2

0∘ , standard parallel 71∘ S) with respect to the WGS84 geoid. X and Y are given in km, and the

270   elevation rate and its standard deviation are given in m/yr.

ICESat & Envisat combination for time period between February 2003 and October

2.5.3   2009.

273   Elevation changes have been corrected for firn densification processes using a firndensification model. The dataset is provided in a 10 km grid in polar stereographic projection

(central meridian 0°, standard parallel 71°∘ S) with respect to the WGS84 geoid. X and Y are

276   given in km, and the elevation rate and its standard deviation are given in m/yr.

2.5.4

*Annual elevation trends from CryoSat-2 derived from a single trend covering the*

*time period 2010-2013.*

279   An acceleration term in areas with dynamic thinning was added to the linear trend to obtain

annual rates. Elevation trends are provided at 10 km resolution in a polar stereographic grid

(central meridian 0°∘ , standard parallel 71°∘S) with respect to the WGS84 geoid. X and Y are

2.5.5

282   given in km and the elevation rate and its standard deviation are given in m/yr.

*Elevation changes from firn model*

Annual firn densification rates over 2003-2013 rates obtained from RACMO2.3. Data is

285   provided in a 27 km polar stereographic grid (central meridian 0∘ , standard parallel 71∘ S) with

respect to the WGS84 geoid.  X and Y are given in km and the annual firn densification rates

in m/yr.

[18]

288    *Snow / ice density map*

[R1_284] To perform the conversion of volume change to mass change, a

 density map is provided in 20 km resolution in a polar

291    2.5.6
stereographic grid (central meridian 0°, standard parallel 71°S) with respect to the WGS84

geoph. X and Y are given in km and density in kg/m³. We provide the data set at this point for

completeness; more details on the generation of this density map is given in Sasgen et al. (2017).

294    *ICESat/Envisat combination mask*

[revised manuscript text omitted]

[R1_370] The systematic differences between Wolstencroft et al. (2015) and the REGINA

393   values for Palmer Land are currently unexplained and a matter of ongoing investigation. For

more detailed analysis of rates and time series at individual sites, see Petrie et al. (in prep b).

[25]

*Table 4. Uplift rates $y^u$ and associated uncertainties $\sigma^u$ (mm/yr) for selected sites where uplift rates are manually evaluated based on the spread of rates obtained by sub-sampling the time series ('rman' method), compared to data published by Thomas et al. (2011), Argus et al. (2014), Wolstencroft et al. (2015). See also 'rman' sites in Table Appendix A.4, Table A.2.*

| Site | REGINA | | Thomas et al. (2011) | | Argus et al. (2014) | | Wolstencroft et al. (2015 ) | |
|------|--------|--------|--------|--------|--------|--------|--------|--------|
| | $y^u$ | $\sigma^u$ | $y^u$ | $\sigma^u$ | $y^u$ | $\sigma^u$ | $y^u$ | $\sigma^u$ |
| bren | **3.1** | 1.1 | **3.9** | 1.6 | **2.1** | 3.7 | **3.2** | 0.8 |
| capf | **4.0** | 1.4 | | | **15.0** | 4.2 | | |
| dav1 | **-1.6** | 0.6 | **-0.9** | 0.5 | **−0.8** | 1.0 | | |
| mait | **0.4** | 1.1 | **0.1** | 0.6 | **1.3** | 0.7 | | |
| mbl3 | **1.3** | 17.9 | **0.1** | 2.0 | | | | |
| bean | **2.1** | 4.3 | | | | | **7.5** | 1.2 |
| gmez | **1.5** | 4.8 | | | | | **5.7** | 0.8 |
| lntk | **4.6** | 3.1 | | | | | **6.0** | 0.7 |
| mkib | **4.7** | 2.6 | | | | | **6.9** | 0.5 |
| trve | **2.5** | 5.6 | | | | | **4.7** | 0.6 |

396

*Data availability*

The GPS data and related code are directly accessible in the Pangaea repository,

3.2    http://hs.pangaea.de/model/Sasgen-etal_2017/In_situ_GPS_uplift_rates.zip

[revised manuscript text omitted]

To define the optimal filter parameters a quadratic sum of the signal corruption and noise

reduction is computed, allowing us to balance both effects, the optimal values are $m_{start} =$ 12 and $n_{pol} = 7$ as indicated in Fig. 4c. These filter parameters are subsequently used. For comparison it is stated that Chambers & Bonin, 2012  found $m_{start} = 15$ and $n_{pol} = 4$  to be optimal for oceanic applications. [R1_511] Note that the signal corruption is assessed only to optimize the de-striping filter. Possible signal degradation due to de-striping is not included in the uncertainty estimate of the optimally filtered GRACE trends. However, signal loss due to the additional smoothing with a 200 km Gaussian filter is accounted for by applying the same filter to the viscoelastic response functions, as well as the altimetry-based input fields (Appendix A.5).

**4.2  Reduction of  non-linear mass variations**

The temporal variations of the Antarctic gravity field show a strong year-to-year fluctuation, apart from the linear trend (Wouters et al. 2014) [R1_476]. A large portion of the non-linear signal in geodetic mass and volume time series is well explained by modelled SMB fluctuations (Sasgen et al. 2010; Horwath et al. 2012). Towards the ultimate goal of isolating the linear GIA signal from time series of mass change, we removed non-linear effects of modelled SMB variations from the GRACE time series; for this we calculate the *monthly cumulative SMB anomalies* with respect to the time period 1979 to 2012 obtained from RACMO2/ANT (Lenaerts et al. 2012).

We then transfer the monthly cumulative SMB anomalies in terms of their water-equivalent height change into the spherical harmonic domain and subtract them from the monthly GRACE coefficients. In principle, the reduction of the SMB variations from the GRACE time interval has two effects: first, it may change the overall gravity field rate derived from GRACE, depending on the assumption of the SMB reference period. Ideally, the reference period reflects

[35]

a state of the ice sheet in which input by SMB equals the outflow by ice discharge, and SMB anomalies estimated for today reflect the SMB component of the mass imbalance. However,

555 any bias in the SMB in the reference period leads to an artificial trend in the ice sheet mass balance attributed to SMB. This is an undesired effect, and to avoid it we de-trend the cumulative SMB time series for the time interval coeval to the GRACE analysis (February 2003

558 to October 2009), before subtracting it from the GRACE gravity fields, rates derived from yielding zero difference in the gravity field rates GRACE (zero difference for before and after processing *Step 2*, (Fig. 3) [R1_402]. The second effect is the reduction of the post-fit RMS

561 residual for this known temporal signal variation. After reducing the SMB variations, the propagated RMS uncertainty of the derived gravity field rate becomes closer to the uncertainty level of the GRACE monthly solutions (Fig. 3).

*Month-dependent weighting*

The quality of GRACE monthly solutions changes with time, for example due to changing orbital sampling patterns (Swenson & Wahr 2006). Fig. 5 shows the temporal evolution of RMS

4.3

[Figure]

*Figure 5 [R1_495].  Residual mass anomaly  of monthly GRACE gravity fields  2003-2011, averaged over the Antarctic region south of −60°S latitude. Shown are results for GFZ RL05a and CSR RL05 and $j_{max} = 50$ and $j_{max} = 90$.*

567

uncertainties of the monthly GRACE gravity fields in the Antarctic region Shown are residual mass anomalies, integrated over Antarctica, after the grid-based removal of the temporal linear

570    trend and annual oscillation components and applying the filtering described in *Step 1* and removing the SMB fluctuations in Step 2 [R1_495]. Note than an annual oscillation component is included to remove possible seasonal fluctuations in SMB not captured by the regional

573 climate model [R1_495]. However, omitting the annual oscillation component yields similar results.-The residual monthly mass anomalies are attributed to noise and are used To improve the accuracy of the estimate of the gravity field rate, we include monthly uncertainties as

576 weights in our least-squares linear regression, applied as Step 3 of the GRACE processing. Fig. 5 shows that these uncertainties are higher during early 2003. Applying the monthly dependent weighting has the effect of reducing the influence of the first months of the year 2003 on the

579 estimated gravity field rate, which is similar to shortening the time series, given the relatively large uncertainties. AlsoAs expected, the post-fit RMS uncertainty associated with the rate reduces, if the early months of the year 2003 are excluded [R1_509]. , indicating that down-

582 weighting the months from early 2003 is more beneficial than retaining a longer time series. Altogether, the month-dependent weighting reduces the magnitude of stripe patterns characteristic for the uncertainty of GRACE monthly solutions, and yields a more accurate

585 realistic representation estimate of the of propagated RMS uncertainty associated with the gravity field rates (Fig. 3) [R1_514].

588    Fig. 6 shows the estimated RMS uncertainty of the gravity field rate over Antarctica, after

post-processing. It is evident that the largest uncertainties are located in a ring south of $-80°$S

latitude.⁴,⁴ This is explained by the design of the Swenson filter; little or no noise reduction is

591   achieved close to the poles, as the gravity field is represented by near-zonal coefficients, which

[Figure]

*Figure 6.  Linear trend in the GRACE gravity fields for the years 2003-2009; a) GFZ RL05a, b)*

*CSR RL05, c) difference between rates from GFZ RL05 and CSR RL05, propagated d) RMS*

*uncertainty for GFZ RL05a and e) RMS uncertainty for CSR RL05.*

pass the filter mostly unchanged ($m_{start} = 12$). Extending the kernel of the

Swenson filter to these near-zonal coefficients ($m_{start} \leq 10$) creates high signal corruption

594   and is not suitable for the optimal rate estimate over Antarctica (see Section 4.1). Larger

uncertainties are also estimated for the Ronne and Ross ice shelf areas, which are most likely a

consequence of incomplete removal of the ocean tide signal during the GRACE de-aliasing

597     procedure (Dobslaw et al. 2013). It should also be stated that the RMS uncertainty estimate

        does not include possible systematic errors in the GRACE solutions, e.g. due to a long-term

        drift behavior of the observing system.

600     *Selection of GRACE release*

        Our evaluation of the monthly GRACE uncertainties (Fig. 5), as well as the propagated

        [45]RMS uncertainty of the temporal linear trend (Fig. 6) indicates that the lowest noise level for

603     the Antarctic gravity field rate (February 2003 to October 2009) is currently achieved with

        GRACE coefficients of CSR RL05, expanded to $j_{max}$ = 50. We therefore refrain from

        including coefficients with $j_{max}$ > 50 in order not to  compromise the rate estimates

606     by unnecessarily increasing the noise level (see Appendix A.5, Fig. A.3). We adopt CSR RL05

        with $j_{max}$ = 50 as our preferred solutions for the representation of the gravity field rates over

        Antarctica, even though GFZ RL05 with $j_{max}$ = 50 yields very similar rates (Fig. 6). This

609     choice is supported by the joint inversion, as CSR RL05 with $j_{max}$ = 50 provides the highest

        level of consistency (lowest residual misfit) with the altimetry and GPS data sets (see REGINA

        Part II, Sasgen et al. 2017, Supplementary Information, Section S.3), which we interpret to

612     indicate  a minimum of spurious signals in the trends. To account for the uncertainty related

        to our choice of the solution, we consider not only RMS uncertainties of the GRACE rates but

        also solution differences, in the uncertainty of the final GIA estimate (Fig. 6). The solution

615     difference represent the absolute deviation between trends from GFZ RL05 and CSR RL05

        (February 2003 to October 2009, cut-off degree $j_{max}$ = 50). These are then summed up squared

        with the propagated RMS uncertainties. It is acknowledged that the solution differences contain

618     systematic noise arising from the GRACE processing; the pattern and magnitude may change

        over time. However, they provide a measure how much the results will change, if a GRACE

release alternative to CSR RL05 is considered. The difference between GRACE rates filtered

621    with Gaussian smoothing of 200 km and the optimized Swenson filter together with Gaussian

smoothing of 200 km is shown in the Appendix A.5, Fig. A.4.

*Data availability*

624    The gravity data and related code are directly accessible in the Pangaea repository,

4.6
http://hs.pangaea.de/model/Sasgen-etal_2017/Geoidheight_change_from_GRACE_satellite.zip

627    *Stokes coefficients of gravity field change*

4.6.1    The monthly GRACE gravity field solutions from the Data System Centers GFZ and CSR

are available under ftp://podaac.jpl.nasa.gov/allData/grace/L2/ or http://isdc.gfz-potsdam.de/ as

630    spherical harmonic (SH) expansion coefficients of the gravitations potential (Stokes

confidents). More information is available in Bettadpur (2012). The data archive contains

temporal linear trends of the fully normalized Stokes coefficients in the 'geodetic norm'

633    (Heiskanen & Moritz, 1967),  complete to degree and order 90, inferred from these time series

according to Section 4. We provide data for GFZ RL05 and CSR RL05, for the time period

2003-2009 and 2003-2013, and for various combinations of filtering. The coefficients are

636    organized as:
4.6.2

[Degree $j$], [Order $m$], [$c\_jm$], [$s\_jm$]

*Code for de-striping filtering*

639    The Matlab® function "KFF_filt" performs decorrelation filtering for sets of spherical

harmonic coefficients, typically from GRACE gravity field solutions, after the idea of Swenson

& Wahr (2006). An open-source alternative to Matlab® is GNU Octave

[41]

642     https://www.gnu.org/software/octave/. The function is called as KFF_filt =

swenson_filter_2(KFF, ord_min, deg_poly, factorvec, maxdeg), where variables ord_min and

deg_poly equal $m_{start}$ and $n_{pol}$, respectively, in Section 4. KFF contains the sets of spherical

645    harmonic coefficients in the 'triangular' format (not memory-efficient but intuitive). For

example, for a set of coefficients with maximum degree $j_{max} = 3$ and maximum order $m_{max} = $

3, the set of coefficients is stored in a $j_{max} \times m_{max}$ matrix in the following way:

648       % KFF = [0    0    0   c_00 0    0    0;

      %    0    0   s_11 c_10 c_11 0    0;

      %    0   s_22 s_21 c_20 c_21 c_22 0;

651       %   s_33 s_32 s_31 c_30 c_31 c_32 c_33]

**5. VISCOELASTIC MODELLING**

654     The Earth structure of Antarctica is characterized by a strong dichotomy between east and

west, separated along the Transantarctic Mountains (e.g. Morelli & Danesi, 2004). Recent

seismic studies have produced refined maps of crustal thicknesses also showing slower upper-

657    mantle seismic velocities in West Antarctica, indicating a thin elastic lithosphere and reduced

mantle viscosity (An et al. 2015; Heeszel et al. 2016). Moreover, yield strength envelopes of

the Earth's crust and mantle suggest the possibility of a viscously deforming layer (DL) in the

660    lower part of the crustal lithosphere (Ranalli & Murphy, 1987), a few tens of km thick and with

viscosities as low as $10^{17}$ Pa s (Schotman et al., 2008). High geothermal heat flux is in

agreement with the seismic inferences of a thin elastic lithosphere and low mantle viscosity,

663    and would favor the presence of such a DL also in West Antarctica (Shapiro & Ritzwoller 2004;

Schroeder et al. 2014).

The choice of the viscoelastic modelling approach used to determine load-induced surface displacements and gravitational perturbations is governed by three main requirements; i) to accommodate a lateral variations in Earth viscosity, ii) to allow for Earth structures with thin elastic lithosphere and low viscosity layers, in particular including a DL, and iii) to provide viscoelastic response functions for the joint inversion of the satellite data described in REGINA paper II (Sasgen et al. 2017 submitted). With regard to point iii) it should be mentioned that the viscoelastic response functions provide a geophysical meaningful way to relate surface displacement and gravity field changes, considering also dynamic density changes within the Earth's interior . Moreover, it allows us to consider the changes in the ratio of surface-displacement and gravity field changes caused by the Earth structure, in particular, the lithosphere thickness. Another advantage is that different filtering can be applied to the viscoelastic response functions in order to match the filtering of the input data set, avoiding the introduction related biases (Appendix A.5)[R1_653].

To meet these requirements, we adopt the time-domain approach (Martinec 2000) for calculating viscoelastic response functions of a Maxell continuum to the forcing exerted by normalized disc-loads of constant radius. Then, the magnitudes and spatial distribution of the surface loads are adjusted according to the satellite data to obtain the full GIA signal for Antarctica. The forward modelling of viscoelastic response functions is a classic topic in sold Earth modelling (e.g. Peltier & Andrews, 1976), however, their application to inverting multiple-satellite observations for present and past ice sheet mass changes is new and applicable to other regions, such as Greenland or Alaska.

The viscoelastic response function approach allows for high spatial resolution at low computational cost in the numerical discretization of the Earth structure as well as in the

[43]

representation of the load and the response. In addition, we can accommodate a high temporal

resolution, which is required when considering low viscosities and associated relaxation times

690    of only a few decades. The spherical harmonic cut-off degree for the simulations shown in the

following is $j_{max} = 2048$ (ca. 10 km).

*Load model parameters*

693    The load function $\sigma(t, \vartheta)$ is disc shaped with a constant radius of ca. 63 km. The radius of

5.1
63 km matches the mean radius of the discs south of 60°S of the geodesic grid (here, ICON 1.2

grid, status 2007, e.g. Wan et al., 2013), which underlie the joint inversion of the altimetry,

696    gravimetry and GPS observations (see REGINA paper II, Sasgen et al. 2017.). The

resolution of the geodesic grid is chosen to allow for an adequate representation of the load and

viscoelastic response with regard to the input data sets, while minimizing the computational

699    cost. The disc load experiment consists of a linear increase in the ice thickness at a rate of 0.5

m/yr continuing until a new dynamic equilibrium state between load and response is reached.

[R2_658] After the application of the constant loading rate, two extra time steps are done with

702    no loading change to give the purely viscoelastic response. For West Antarctica, the loading

rate is held constant for 2000 years, for East Antarctica it is 15,000 years, which are longer

times than needed to reach dynamic equilibrium (see Appendix A.8). With reference to the

705    assumed ice density of 910 kg/m³, this thickness increase corresponds to a mass gain of *ca.* 5.6

Gt/yr. Then, to obtain the signal component of the viscous Earth response only, the elastic

response and the direct gravitational attraction of the load are subtracted.

708    The experiment is designed as an *increasing* load, for example representative for the

ceasing motion of the Kamb Ice Stream (Ice Stream C; Retzlaff & Bentley, 1993), West

Antarctica. Due to linearity of the viscoelastic field equations, it is not necessary to calculate

[44]

711      separately the equivalent *unloading* experiment, $-\sigma(t, \vartheta)$, for example corresponding to the

past and present glacier retreat of the Amundsen Sea Sector, West Antarctica (Bentley et al.

2014 and Rignot et al. 2014, respectively). Among others, the combined inversion of the

714      altimetry, gravity and GPS data (REGINA paper II, Sasgen et al. 2017) solves for the magnitude

and the sign of the load, allowing for ice advance as well as ice retreat.

*Earth model parameters*

717      We set up an ensemble of 58 simulations representing different parameterizations of the

5.2

viscosity structure (Table 5), split into West Antarctica (56 simulations) and East Antarctica (2

simulations). The ensemble approximately covers the range of values of the viscosity and

720      lithosphere thickness inferred from Priestley & McKenzie (2013) [R1_696]. For West

Antarctica, varied parameters are the lithosphere thickness, $h_L$ (30 to 90 km in steps of 10 km),

the asthenosphere viscosity ($1 \times 10^{18}$ Pa s to $3 \times 10^{19}$ Pa s in four steps), and the presence of

723      a ductile lower crust, DL, with $10^{18}$ Pa s. For East Antarctica, we employ parameter

combinations appropriate for its cratonic origin with $h_L$ of 150 km and 200 km, and an

asthenosphere viscosity equivalent to the upper-mantle viscosity of $5 \times 10^{20}$ Pa s. These values

726      lie in the range of previously applied viscosity values in Antarctica (Nield et al. 2012;

Whitehouse et al., 2012; Ivins et al., 2013; van der Wal et al., 2015). For the radial layering of

the elastic properties, we adopt the Preliminary Reference Earth Model (PREM; Dziewonski &

729      Anderson 1981).

*Table 5. Earth model parameters associated with the disc load ensemble simulations. The viscoelastic parameterization of the Earth model is discretized in six radial layers; upper and lower crust, mantle lithosphere, asthenosphere, upper and lower mantle. The lower mantle extends down to the core mantle boundary (CMB; at the depth of 2763 km). Elastic layers are represented by a quasi-infinite viscosity of $10^{30}$ Pa s.*

| Layer | Depth (km) | Viscosity (Pa s) | Unique param. val. |
|---|---|---|---|
| **West Antarctica** | | | |
| Upper crust | 20 | $10^{30}$ | 1 |
| Lower crust DL [yes/no] | 30 | $[10^{30}/10^{18}]$ | 2 |
| Mantle lithosphere | [30, 90, steps of 10] | $10^{30}$ | 7 |
| Asthenosphere | 200 | $[1\times10^{18}, 3\times10^{18}, 1\times10^{19}, 3\times10^{19}]$ | 4 |
| Upper mantle | 670 | $5\times10^{20}$ | 1 |
| Lower mantle | CMB | $2\times10^{22}$ | 1 |
| Number of simulations West Antarctica | | | 56 |
| **East Antarctica** | | | |
| Crust | 30 | $10^{30}$ | 1 |
| Mantle lithosphere | [150, 200] | $10^{30}$ | 2 |
| Upper mantle | 670 | $5\times10^{20}$ | 1 |
| Lower mantle to CMB | CMB | $2\times10^{22}$ | 1 |
| Number of simulations East Antarctica | | | **2** |
| **Elastic earth** | | | |
| Crust and mantle to CMB | CMB | $10^{30}$ | 1 |
| **Total number of simulations** | | | 59 |

Later, in the joint inversion, the distribution of viscoelastic response functions is based on the Earth structure model of Priestley & McKenzie (2013). Priestley & McKenzie (2013) provide a global distribution of viscosity values up to a depth of 400 km, which is sampled at the location of the geodesic grid. We then define a threshold value for the viscosity (here, 10$^{22}$ Pas) above which the Earth response is considered purely elastic and infer the associated thickness of the elastic lithosphere. The impact on the final joint inversion estimate of changing

[46]

the threshold value of 10 $^{22}$ Pas is presented in REGINA Paper II (Fig. S4 in Sasgen et al. 2017) [R1_712]. 
[revised manuscript text omitted]

functions describe the Earth response in an equilibrium state for a constant rate of load change;

if the load exhibits more complex temporal variations, this assumption is violated. Finally, it is

819  assumed that the lithosphere thickness, upper- and lower-mantle viscosities are approximately

known. [R1_795].

[53]

*Data availability*

The viscoelastic response functions and related ancillary data are directly accessible in the

Pangaea repository:

http://hs.pangaea.de/model/Sasgen-etal_2017/Viscoelastic_response_functions.zip

[revised manuscript text omitted]

to the GPS uplift rates given that the time spans of these data vary [R2_867].

We have optimized the post-processing sequence for estimating the temporal linear trend

921 and its uncertainty in the GRACE gravity field solutions for the region of Antarctica. In

particular, we have derived optimal parameters for de-striping the monthly gravity fields over

Antarctica according to Swenson & Wahr (2006). In addition, we have removed de-trended

924 interannual SMB fluctuations [R1_476] from the GRACE time series, to obtain a more

representative uncertainty estimate based on the post-fit RMS residual. We have included

month-dependent weighting in the least-squares estimate of the gravity field rates to account

927 for the varying quality of the monthly GRACE solutions. The optimization of the de-correlation

filter of Swenson & Wahr (2006) to the signals expected in Antarctica reduced the residual

uncertainty and improved the reliability of inferred mass anomalies.

930 With the aim of joining the multiple satellite data using the knowledge of the geophysical

processes involved, we have calculated elastic and viscoelastic response functions of the solid

Earth. The viscoelastic response functions represent the gravity field change and surface

933 displacement to a disc-load forcing for a variety of Earth model parameters; particularly,

however, values of mantle viscosity and lithosphere thickness strongly varying between the

distinct geological regimes of West and East Antarctica.

936 In particular, we have investigated the effect of a ductile layer in the crustal lithosphere on

the viscoelastic rebound signature. We show that for moderate load changes of 0.45 m/yr water-equivalent (here, applied as disc load with a radius of ca. 63 km), uplift rates reach the cm/yr level within decades assuming asthenosphere viscosities $< 10^{19}$ Pa s and lithosphere thickness $< 50$ km; both plausible values for parts of West Antarctica. Including a ductile layer in the crustal lithosphere further attenuates the uplift rates and localizes the deformational response. This suggests that GIA in West Antarctica may locally be a result of more recent, centennial load changes, most notably in the Amundsen Sea Embayment and in part of the Antarctic Peninsula (Nield et al. 2012). Similar conclusion were reached by Ivins & James (2005) and Nield et al. (2014), even though it is not possible to constrain the exact timing of the load from our approach [R1_921].

The advantage of the viscoelastic response kernels is that a meaningful ratio of rate of the gravity disturbance versus rate of the surface displacement [R1_923] is calculated for each choice of the Earth model parameters, avoiding the approximation with an average rock density (e.g. Riva et al. 2009; Gunter et al. 2014). Using the response functions allows us to reconcile GIA signatures with measurements of large bedrock uplift and small gravity field increase in the Amundsen Sea Embayment, associated with weak Earth structures. Clearly, the response functions adopted here represent only the viscoelastic equilibrium state and, thus, are considered only an intermediate step to full dynamic modelling of the GIA response. Nevertheless, this approximation represents a significant improvement of other joint inversion methods, as it bases the joint inversion on physically meaningful response kernels. With extra data on the past ice evolution, such as Paleo thickness rates, our approach can be expanded to address the temporal evolution as well.

In the succeeding paper REGINA part II (Sasgen et al. 2017submitted), we perform the

960 joint inversion for present-day ice-mass changes and GIA in Antarctica, based on the input data sets and viscoelastic response functions presented here. We validate our results using forward-modelling results and other empirical models, and show the impact on CryoSat-2 volume and

[revised manuscript text omitted]

A.7 *Assessment of SMB fluctuations on GPS uplift rates* [R2_867]

We assess the impact of SMB fluctuations on the uplift rate at the GPS station locations

[Figure]

*Figure A,6: Standard deviation (2-sigma) of the uplift rates caused by accumulation variability for different GPS stations and time periods. It is visible that the uncertainty decreases with record length; for most regions, trend uncertainties are below 0.4 mm/yr for the actual GPS record length.*

using the modelled SMB of RACMO2 for the years 1979-2010. We compute the elastic

deformation related to cumulative monthly SMB, de-trended for the entire simulation period

1053 1979-2010. We then estimate the temporal linear trends at the GPS station locations for a

moving window of varying width from 3 to 16 years. Then, for each window width, we estimate

the standard deviation of the apparent trend induced by SMB for selected stations (Fig. A.7).

1056 Typically, the uncertainty of uplift rate due to SMB variability is below 0.4 mm/yr for the actual

GPS record length. An exception is PALM, which is located on the Antarctica Peninsula - a

region with annual accumulation of up to 4 m/yr equivalent water height. Here, even after 12

1059 years of measurements, GPS uplift rates are likely to contain accumulation signals of 4 mm/yr.

A similar effect of the SMB fluctuations is expected at VESL.

**A.8      *Load evolution for the viscoelastic response functions [R2_867]**

1062      The load increases, with a fixed radius, at a constant rate of *ca.* 5.6 Gt/yr until an

approximate equilibrium state is reached; 2 kyr for West Antarctica and 15 kyr for East

Antarctica (Fig. A.7). Then the load is applied without a change to obtain the purely viscoelastic

1065 response of the Earth model, i.e. without direct gravitational attraction of the load and the

instantaneous elastic response. The associated Earth response constitutes the viscoelastic

response functions adopted in the joint inversion.

[Figure]

1068

[R2_1014] *Figure A,7: Load function applied to obtain the viscoelastic response functions.*

[revised manuscript text omitted]

---

## Author Comment (AC2) · 12 Nov 2017

Dear Reviewer 2,

Thank you very much for your constructive suggestions on how to improve our manuscript.

We have now addressed all your concerns and modified the manuscript accordingly. Please find as a *.pdf supplment our point-by-point Reply and a modeified Version of the ms; the labels in brackets refer to the position in the text.

[Figure]

We also very much appreciated you commented printouts attached as a pdf supplement, which made it very easy for us to include your minor corrections.

Please let us know if you have further comments that need resolving.

Sincerely, Ingo Sasgen & the REGINA team

Please also note the supplement to this comment:
https://www.earth-syst-sci-data-discuss.net/essd-2017-46/essd-2017-46-AC2-supplement.pdf